# Bacteria and Bacterial Components as Natural Bio-Nanocarriers for Drug and Gene Delivery Systems in Cancer Therapy

**DOI:** 10.3390/pharmaceutics15102490

**Published:** 2023-10-19

**Authors:** Rui Zong, Hainan Ruan, Chanmin Liu, Shaohua Fan, Jun Li

**Affiliations:** School of Life Science, Jiangsu Normal University, Xuzhou 221116, China

**Keywords:** bacteria, bacterial components, biomimetic nanoplatforms, tumor targeting, gene therapy, engineered strategies

## Abstract

Bacteria and bacterial components possess multifunctional properties, making them attractive natural bio-nanocarriers for cancer diagnosis and targeted treatment. The inherent tropic and motile nature of bacteria allows them to grow and colonize in hypoxic tumor microenvironments more readily than conventional therapeutic agents and other nanomedicines. However, concerns over biosafety, limited antitumor efficiency, and unclear tumor-targeting mechanisms have restricted the clinical translation and application of natural bio-nanocarriers based on bacteria and bacterial components. Fortunately, bacterial therapies combined with engineering strategies and nanotechnology may be able to reverse a number of challenges for bacterial/bacterial component-based cancer biotherapies. Meanwhile, the combined strategies tend to enhance the versatility of bionanoplasmic nanoplatforms to improve biosafety and inhibit tumorigenesis and metastasis. This review summarizes the advantages and challenges of bacteria and bacterial components in cancer therapy, outlines combinatorial strategies for nanocarriers and bacterial/bacterial components, and discusses their clinical applications.

## 1. Introduction

Cancer remains one of the most threatening diseases that compromise human health and safety, and recent studies suggest that the number of people suffering from cancer worldwide could increase from 19.3 million to 28.4 million between 2020 and 2040 [1]. As a result, more innovative and aggressive strategies are needed to break through the oncology treatment dilemma. Drug delivery has emerged as one of the 100 disruptive technologies of the 21st century, and nano drug delivery systems (nano DDS) have already undergone a clear revolution in this field as an important ingredient. Conventional nanoparticles (NPs) are classified as polymeric NPs, inorganic NPs, and lipid-based NPs [2,3]. Therapeutic nanomedicines have tremendous advantages in many areas that threaten human medical health, such as being antibacterial, antitumor, and anti-infection [4,5,6,7,8]. The development of nanomaterials is beneficial to overcome the limitations of conventional therapies, such as non-uniform biodistribution of drugs, poor intracellular targeting to specific organelles, and short circulation time in vivo [2,3]. Nanocarriers with tunable size enhance drug penetration through high permeability and retention effect (EPR) [9], which can be additionally improved via modifying tumor-targeting ligands in deep solid tumor tissues [10,11,12]. However, the recognition and clearance of nanocarriers by the immune system becomes one of the biggest obstacles to clinical translation. Therefore, the application of polyethylene glycol (PEG) to coat the surface of nanodrugs, or “PEGylation”, can significantly improve the drug circulation time and tumor penetration in vivo [13,14,15] and protect nanodrugs from degradation and clearance by enzymes and antibodies [2,16]. In addition, high concentrations of PEGylation may lead to rapid clearance of NPs due to the generation of anti-PEG antibodies in human and mouse models [16]. In addition, artificial nanomaterials seem to be highly limited in terms of clinical translational applications because of their complex composition, leading to less controllable quality, high cost limiting mass production [17], uncertain biocompatibility, high off-target effects, and rapid drug blood clearance [10]. Upon analysis, it was found that the aggregate quantity of clinical trials registered on clinicaltrials.gov (via the utilization of search terms ‘cancer’, ‘liposome’, and ‘nanoparticle’) and the quantity of publications retrievable via a search on PubMed (via the search terms ‘cancer’ and ‘nanoparticle’) for May 2023 only account for a mere 3% of the total publications available. This phenomenon illustrates the huge gap between the continuous development of nanotechnology and the clinical translation of nanomedicines. Therefore, there is an urgent need for researchers to advance the development of novel and precisely targeted therapeutic strategies for the next generation of clinical nanomedicines.

Engineered bio-nanocarriers have shown great potential to improve the efficacy of cancer diagnosis and tumor-targeted precision therapy [2,14]. Tumor therapy based on bacterial/bacterial components may be an excellent strategy to induce systemic antitumor immune responses, stimulating the immune system by remodeling the tumor microenvironment (TME) and inhibiting tumor cell proliferation and metastasis. The TME, composed mainly of tumor cells, a vascular system, tumor-associated fibroblasts, and immune cells, is a crucial factor that impacts cancer treatment. The complexity of the TME is one of the key factors affecting cancer progression [18,19,20]. Tumor cells, vasculature, tumor-associated fibroblasts, and immune cells are the main components that constitute the TME [21,22]. The complex and heterogeneous TME not only facilitates the growth and metastasis of solid tumors, but also helps tumor cells to achieve immune evasion and generate multidrug resistance, which severely hinders the progress of tumor therapy [21,23]. However, TME contributes to the hypoxic, low pH, and elevated interstitial fluid pressure characteristics of solid tumor sites [10,24], which favors the growth and colonization of anaerobic and facultative anaerobic bacteria [25]. Simultaneously, bacteria with inherent tumor-targeting capacities tend to aggregate in the anaerobic and low-pH tumor microenvironment. The antitumor efficacy of bacteria has been known since the 19th century [10]. As shown in Figure 1A, we present typical cases of bacterial-mediated cancer therapy from 1893 to 2022. The *Bacillus Calmette Guerin* (BCG), derived from attenuated *Mycobacterium tuberculosis* [26], has been clinically proven to be an excellent anticancer agent that effectively inhibits the growth and metastasis of solid tumors, such as lung, prostate, colon, bladder, and kidney cancers. It has become the only FDA-approved drug for the treatment of non-muscle-invasive bladder cancer [19]. Meanwhile, *Salmonella*, *Clostridium*, and *Listeria* have been shown to inhibit tumor progression in mouse models and clinical trials [19]. Due to the rapid advances in TME and tumor microbiome research in recent years [18], novel strategies for tumor treatment with bacteria and bacterial components have attracted more attention and research (Figure 1B).

Bacteria can be natural drug carriers, delivering agents and therapeutic drugs for application in cancer therapy [27,28]. Bacterial components, mainly including bacterial outer membrane vesicles (OMVs), bacterial ghosts (BGs), bacterial spores (BSPs), and others, act as immunostimulatory adjuvants to generate potent antitumor immune responses and modulate TME in cancer therapy [10,21,29,30]. These significant advantages make bacterial/bacterial component-based drug carriers a potential strategy for the treatment of cancer. Although bacteria and bacterial components are used as bionic nanocarriers for drug delivery due to their excellent tumor-targeting capabilities, concerns about bacterial pathogenicity and the emergence of numerous challenges, such as low efficacy, have greatly hindered the progress of bacterial-based biotherapies [30,31]. Therefore, engineering strategies and nanotechnology combined with bacterial/bacterial components may be an effective breakthrough to improve and enable multifunctional bacterial cancer therapy [20,32].

This review highlights recent advances in natural bio-nanocarriers based on bacteria and bacterial components as drug and gene delivery systems, and elucidates the advantages and challenges of bacteria and bacterial components in cancer therapy. We discuss the combination of bacteria/bacterial components and nanotechnology to improve the antitumor efficacy and biocompatibility of bacteria-based nanoplatforms. In addition, this article describes the current challenges, future trends, and directions of bacteria and bacterial components as bio-nanocarriers in cancer therapy, as well as their clinical applications.

## 2. Bacteria and Bacterial Components in Cancer Therapy

### 2.1. Bacteria in Cancer Therapy

Bacteria-mediated tumor biotherapy is emerging as a promising approach in cancer treatment compared to conventional cancer therapies [19,32]. Interestingly, anaerobic or facultative anaerobic bacteria could be attracted into hypoxic and low-pH-specific vascular composition and the necrotic region in the TME [10,21]. Thus, obligate or facultative anaerobic microorganisms will selectively congregate and proliferate in the hypoxic regions of the tumor. Several clinical trials have shown that anaerobic bacteria can promote tumor regression and inhibit cancer metastasis in a variety of ways, including toxin production, the activation of T-cell immune responses, and synergistic effects with therapeutic agents [33,34,35]. Obligate anaerobic bacteria such as *Clostridium* and *Bifidobacterium* have difficulty surviving in the oxygen-rich zone of normal cells, and they tend to migrate into the hypoxic TME [19]. The inherent mobility of such living bacteria allows them to overcome the intravascular diffusion resistance of solid tumors and invade the deeper domains of cancer compared to conventional chemotherapeutic drugs and traditional nanomedicines. Thus, bacterial-mediated drug delivery systems are effective therapeutic strategies that may eradicate solid tumors due to their highly tropic and motile nature. Live bacteria tend to migrate, grow, and colonize into TME that are hypoxic and low-pH, and strong immunogenicity may lead to biosafety concerns. Probiotics, including *Escherichia coli* (*E. coli*) Nissle 1917, *Saccharomyces boulardii* (*S. boulardii*), *Lactobacillus reuteri* (*L. reuteri*), *Lactobacillus casei* (*L. casei*), *Lactobacillus rhamnosus* (*L. rhamnosus*), *Bifidobacterium infantis* (*B. infantis*), and *Bifidobacterium breve* (*B. breve*), can directly target tumor sites with few adverse side effects, including bacterial infection or inflammation [36,37]. Although *E. coli* [38,39,40], *Clostridium* [41], *Listeria monocytogenes*, and *Salmonella typhimurium* [42,43] have shown significant effects on cancer remission, nonpathogenic and genetically modified attenuated bacteria modulate TME at best and do not produce superior efficacy at solid tumor sites [29]. Thus, the challenge of bacterial therapy in cancer will remain daunting in the future.

### 2.2. Bacterial Components in Cancer Therapy

Bacterial components mainly consist of bacterial OMVs, BGs, BSPs, bacterial proteins, and bacterial endopolymers [44], and OMVs and BSPs are considered as potential biomimetic nanomaterials with the potential to act synergistically, with EPR effects of macromolecular substances and lipid particles of a certain size possessing selective enhanced permeability and retention in solid tumor tissues [30,45,46]. More importantly, bacterial components can be recognized by the host immune system and trigger antitumor immune responses [47].

#### 2.2.1. Bacterial Outer Membrane Vesicles (OMVs)

Bacterial OMVs are nanosized spherical lipid bilayer vesicles containing genetic material (DNA and RNA) and some metabolites [30,48,49]. The particle size of bacterial OMVs (20–250 nm) is consistent with the EPR effects to target tumors. Bacterial OMVs can be released from both Gram-negative and Gram-positive bacteria (Figure 2), and the main component of the outer membrane of Gram-negative bacteria is lipopolysaccharide (LPS) known as endotoxin [17,50]. The two kinds of bacterial OMVs differ from one another in numerous ways, including the presence of LPS, specific internal proteins, and metabolites [30,51]. 

Explosive lysis and cell membrane blebbing are the two main mechanisms for the production of Gram-negative bacterial OMVs [51]. Cell membrane detachment leads to the production of classical bacterial OMVs, which contain outer membrane proteins and specific lipid components. In this mechanism, the inner membrane is intact and the cytoplasmic component could not theoretically be contained in the OMVs [51]. Therefore, the presence of cytoplasmic content within classical OMVs needs to be further investigated [52]. In contrast, in other important mechanisms, endolysin-induced explosive lysis of bacteria disrupts the intracellular membrane, producing explosive outer membrane vesicles (EOMVs) and outer inner membrane vesicles (OIMVs), both of which contain cytoplasmic components. The former contains a single outer membrane, while the latter contains a double-membrane bilayer (Figure 2).

On the other hand, cytoplasmic membrane vesicles (CMVs) are generated in Gram-positive bacteria as a result of surface bleeding and bacterial death [30,51]. CMVs contain both membrane and cytoplasmic ingredients. The thicker cell walls of Gram-positive bacteria with render it more difficult for them to produce extracellular vesicles than Gram-negative bacteria. The release of bacterial OMVs is influenced by several factors, such as temperature, medium mixture, pH value, oxygen content, and antibiotics [53,54]. Bacterial OMVs possess the same ability to encapsulate active ingredients as liposomes, and are biomimetic materials with great biocompatibility [17,55]. However, the surface of bacterial OMVs can be easily modified, while the unique antigens and pathogen-associated molecular patterns (PAMPs) found in bacterial OMVs, including LPS, peptidoglycan, and flagellin, can enhance immune synergism [17,23,39,56,57]. PAMPs stimulate dendritic (DC) cells by mediating T-cell responses, resulting in superior tumor-killing efficacy [58,59]. Thus, OMVs may possess superior biomedical value in tumor immunotherapy compared to liposomes.

Figure 3 shows the classical methods for the isolation and purification of bacterial OMVs. Briefly, the isolation and purification of bacterial OMVs are investigated through a series of centrifugation and filtration steps. Specifically, the bacterial medium is subjected to low-speed centrifugation to eliminate bacterial cells and debris [60], followed by sequential filtration to remove residual bacteria. The resulting supernatant containing OMVs and other components is then concentrated via a hollow fiber membrane to remove non-OMV components [61,62], which is usually enriched via ultrafiltration [63]. Hereafter, the retentates are subjected to ultracentrifugation to obtain the primary extract of unpurified OMVs. The primary extract of OMVs is applied for experiments in many studies [64,65,66,67,68]. But the primary extract contained many other components such as flagella, large protein complexes, and some small molecules that are not conducive to the development of subsequent in-depth experiments [30,69]. Subsequently, ultracentrifugation is performed to obtain the primary extract of unpurified OMVs, and then Optiprep/sucrose density gradient ultracentrifugation is applied to further enrich and purify the OMVs [70,71]. Overall, numerous studies provide evidence that ultrafiltration and ultracentrifugation are effective methods for the isolation and purification of bacterial OMVs [72,73,74].

However, the popularity of this extraction method is greatly limited by its low extraction and purification efficiency. In a pioneering study, a method based on the broad-spectrum antimicrobial agent epsilon-poly-L-lysine was developed, which has a strong positive charge and can bind to negatively charged bacterial OMVs by electrostatic adsorption [75]. The authors found that OMVs isolated from *E. coli* DH5α and *S. aureus* CICC 10,384 exhibited similar size, protein content, and immunogenicity compared to conventional ultracentrifugation methods. This study provides a more feasible approach with the advantages of simpler operation, easier access, lower cost, and greater yield. In another example, Park et al. [40] introduced a simple method to produce artificial OMVs based on ultrasound, lysozyme, and high-pH treatment. The size and morphology of artificial OMVs were similar to those of natural OMVs. Interestingly, artificial OMVs were 15-fold purer and produced 40-fold higher concentrations compared to the same volume of natural OMVs.

Bacterial OMVs of uniform size from *Bordetella bronchiseptica* were isolated by ultrasonic methods [76]. Artificial OMVs had significantly lower expression levels of the pro-inflammatory cytokine IL-6 than conventional bacterial OMVs, demonstrating the better biosafety of artificial OMVs. Similarly, artificial OMV vaccines can stimulate internal immune responses. In another study, bacterial pellets of the nonpathogenic strain *Mycobacterium smegmatis* were co-cultured with chemical reagents (Tris-HCl and SDS), then the bacteria were lysed in a chloroform–methanol–water solution and the lysate was passed through Millipore membranes [77]. The pellets were vacuum-dried and finally dissolved in PBS to obtain bacterial OMVs with uniform size and similar zeta potential. A recent study showed that the bacterial pellets from *E. coli* were subjected to ultrasonic cell disruption [78]. The pellets of bacterial OMVs can be obtained by freeze-drying after high-speed centrifugation to remove the precipitate. Additionally, binding resin and OMV supernatant might be employed together for separation and purification with decent OMV purity [79]. In a unique study, Gao et al. successfully disrupted microorganisms using nitrogen cavitation vessels, achieving impressive yields [22]. From this perspective, classical ultracentrifugation and ultrafiltration are not the only methods for preparing bacterial OMVs due to their low yields. Ultrasonic disruption or nitrogen cavitation combined with chemical reagents could produce bacterial OMVs with comparable functionality, homogeneous size, and higher yields. Therefore, various studies provide remarkably innovative directions for the advancement of bacterial OMV-based research.

#### 2.2.2. Bacterial Ghosts (BGs)

BGs originating from Gram-negative bacteria are considered bacterial envelopes that do not contain all the cytoplasm and contents of the microorganism, which contain PAMPs and other components on the surface of bacteria [80]. Since the bacterial cell membrane is completely preserved without any denaturation, BGs possess the same ability to stimulate immune responses as live bacteria. Therefore, BGs are one of the most studied bacterial component-based vehicles used as anticancer vaccines for tumor-targeting strategies [81,82].

Figure 4 demonstrates that genetic engineering and chemical methods have become the two predominant strategies for the preparation of BGs. The most prevalent approach for generating BGs in Gram-negative bacteria involves transfecting the lysis gene *E* [83,84], which inhibits the synthesis of hydrophobic proteins in the bacterial cell wall and increases osmotic pressure, resulting in the leakage of bacterial contents [81]. Xie et al. [83] proposed a technique of transfecting phage-derived lysis gene *E* into *E. coli* Nissle 1917 (EcN) using electroporation. The expression of the EcN-induced lysis gene *E* was observed, and antibiotics were added to deactivate the unlysed bacteria. The successful transformation of BGs is characterized by a sudden decrease in the OD_600_ value of the bacteria after incubation, accompanied by the emergence of nanosized pores on the bacterial surface, indicating the outflow of bacterial contents. Similarly, high-quality BGs through the transfection of lysis gene *E* via electroporation are also reported [84].

In another approach, BGs were prepared using chemical reagents and co-cultured bacterial cell walls by determining the minimum inhibitory concentration (MIC) or minimum growth concentration (MGC) [85,86,87,88]. Ji et al. co-incubated *Listeria monocytogenes* (*L. m*) with HCl, H_2_SO_4_, and NaOH at the MIC to create lysis pores on the surface of the bacterial membrane [86]. The resulting *L. m* BGs were found to activate cellular and humoral-mediated immune responses, showing potential as vaccines. Two types of BGs were derived from *Kocuria* sp., including living bacterial ghosts (L-BGs) and dead bacterial ghosts (D-BGs) [87]. L-BGs were prepared by heating in water at 40 °C followed by centrifugation through a 0.22 pm PVDF membrane. In contrast, D-BGs were incubated with SDS, CaCO_3_, and NaOH, followed by centrifugation, and the precipitate was resuspended in 60% ethanol to remove organic matter. In another study, *Salmonella enterica serovar typhimurium* (ATCC 13311) was incubated with 7% *v*/*v* Tween 80, followed by the addition of pH 3.6 lactic acid [85]. High-speed centrifugation separated the undamaged bacterial pellets, and the resulting BGs exhibited intact bacterial shells and intra-membrane pores determined by scanning electron microscopy. This protocol is relatively simple and economical compared to genetic engineering and other chemical methods and can be applied to both Gram-negative and Gram-positive bacteria. However, it should be noted that the chemical method may cause the denaturation of functional proteins on the surface of the bacterial membrane.

Although genetic engineering is a practical way to transform most Gram-negative bacteria with the lysis gene *E*, certain issues still need to be resolved. For instance, the uneven distribution of the lysis gene *E* and competition with antibiotic resistance genes during bacterial division can be problematic. Furthermore, *E. coli* can be resistant to cleavage mutations. The use of chemical reagents to prepare BGs is relatively simple and rapid, and it can be applied to both Gram-negative and Gram-positive bacteria. However, it may cause the denaturation of functional proteins on the bacterial membrane surface.

#### 2.2.3. Bacterial Spores (BSPs)

BSPs are dormant life forms of probiotics that can become metabolically active vegetative cells upon disintegration of their hydrophobic coat [25,89]. BSPs have recently been utilized as a drug delivery platform and have shown promising results in enhancing antitumor vaccine efficacy [90,91]. *Clostridium Novyi-NT* spores, which are widely employed, have been confirmed to lack deadly toxins and are suitable for surviving in anaerobic and low-pH environments found in solid tumors [92]. The extraction process of *Clostridium Novyi-NT* involves growth in anaerobic conditions at 37 °C for seven or ten days, followed by sorting and purification using density gradient centrifugation [93]. Clinical trials have demonstrated effective antitumor effects of *Clostridium Novyi-NT* spores [41]. However, its severe immune response and limited therapeutic potential remains a challenge that requires further investigation.

#### 2.2.4. Other Bacterial Components

Bacterial polymers, encompassing polysaccharides, polyamides, poly (γ-glutamic acid), polyphenols, polyesters, and hyaluronic acid, represent other prominent constituents of microorganisms that offer promising opportunities for the development of bacteria-derived agents in nanomaterial synthesis. These bacterial polymers hold immense potential in enhancing controlled drug release and osmotic efficacy within the realm of nanomedicine [44]. Bacterial outer membrane proteins derived from *Klebsiella pneumonia* and *Neisseria meningitides* have been proven effective as immune adjuvants to activate the immune system for cancer therapy in clinical trials [94,95].

### 2.3. Advantages and Challenges of Bacteria-/Bacterial Component-Based Delivery Vector

Bacteria and their components have emerged as potential drug delivery systems due to their intrinsic properties. These properties enable them to deliver therapeutic medicines to disease sites, protect biologically active molecules from degradation, and enhance the stability of encapsulated drugs. Compared to traditional medicines, bacterial therapy offers several advantages, including high drug delivery efficiency, intrinsic immunostimulatory properties [10], easy modification [25], and efficient tumor-targeting ability [27,30] (Figure 5). However, the clinical translation of bacteria and bacterial components as drug delivery systems is currently limited due to potential toxicity and immunogenicity [25], uncontrollable drug dose and concentration [38], ambiguous mechanism [10,19], and low therapeutic efficiency [30]. 

To overcome these limitations, personalized and precise therapeutic options can be achieved through engineered tactics and the combination of nanotechnology. Engineered strategies, such as genetic engineering, can be employed to weaken the virulence genes of bacteria and their components [53,96]. Additionally, bacteria and bacterial components can be designed with multiple functions to increase therapeutic effectiveness [27]. The implementation of these strategies in combination with nanotechnology could facilitate the clinical translation of bacteria-/bacterial component-based drug delivery systems. In summary, the intrinsic properties of bacteria and bacterial components make them a promising drug delivery system. However, the limitations associated with their use must be addressed through engineered strategies and nanotechnology to realize their full potential in clinical translation.

## 3. Engineering Strategy-Based Bacteria and Bacterial Components for Cancer Therapy

### 3.1. Engineering Bacteria for Drug Delivery in Cancer Therapy

Bacteria are applied as bionic drug carriers for tumor treatment due to their inherent versatile properties. However, the worried of biosafety and limited therapeutic efficiency hindered the advances of bacteria-mediated antitumor therapy. Therefore, the combination of bacteria and engineering strategies can bring out synergetic therapeutic. Common methods of combining bacteria with engineering strategies are as follows (Table 1 and Figure 6): chemical binding, genetic engineering, and biomimetic cell-surface coating in order to produce more splendid tumor-targeting therapeutic effects.

#### 3.1.1. Chemical Binding

As biomimetic antitumor carriers, bacteria often combine with other anticancer drugs through chemical bonds such as amination reaction and polydopamine. For example, attenuated *Salmonella typhimurium* VNP20009 was employed for the intratumoral delivery of living bacterial drugs, which was covalently linked with nucleic acid aptamer through one-step amination reaction for the specific recognition of solid tumors [38]. This study showed that the localization ability of the bacteria in the tumor sites of H22 liver cancer was increased and showed excellent suppression of tumor sites after chemical binding. Recently, a genetic engineering and polymer chemistry technique was reported to modify *E. coli* BL21 for producing melanin with photothermal effects [97]. Anti-programmed death-1 (αPD-1) antibody as a kind of immune checkpoint inhibitor was attached to the surface of *E. coli* by polydopamine. The attachment of chemical binding had no significant effect on the viability of the living bacteria. Therefore, the engineered bacteria remained able to target and colonize the anaerobic tumor environment. Internally encapsulated melanin of *E. coli* was able to release melanin in the solid tumor field of 4T1 tumor-bearing mice models under NIR irradiation. The proportion of CD80^+^ and CD86^+^ mature DC cells in mice was increased after administration and the immune response of T cells was stimulated upon photothermal therapy (PTT). Similarly, the production of IFN-γ secreted by activated T cells was increased. The distribution and retention of αPD-1 further enhanced the ability of dual tumor immune activation at the tumor sites, significantly inhibited tumor growth, and improved survival in 4T1 tumor-bearing mice.

#### 3.1.2. Genetic Engineering

Numerous studies have combined genetic engineering with bacterial cancer therapy, and the most direct purpose is to remove the virulence genes of pathogenic bacteria and improve biosafety. For instance, the removal of the *msbA*/*msbB* gene could effectively reduce the adverse infection and inflammation caused by bacteria [105,106,107]. The deletion of the *msbB* gene in bacterial strains results in myristoylation of the lipid A component of LPS, leading to a significant reduction in the risk of septic shock [108]. This method of genetic engineering has demonstrated the capability to decrease the toxicity of *Salmonella typhimurium* VNP20009 by 10,000-fold [109], exhibiting enhanced accumulation at the tumor site with a tumor-to-liver colonization ratio of 1000:1. Clinical trials (NCT00004988, NCT04589234) involving attenuated *Salmonella typhimurium* for cancer therapy have indicated minimal side effects. An alternative strategy for enhancing the safety of bacterial cancer therapy involves generating nutrient-deficient mutants. These mutants exhibit selective enrichment at the tumor site, unable to replicate or spread in normal tissues, representing a promising avenue for future bacterial tumor vaccines [109,110].

Certainly, researchers expect that the combination of bacteria with functional genetic engineering strategies could yield more excellent antitumor efficiency. For example, attenuated *Salmonella typhimurium* was used in combined with photodynamic therapy, which was transformed with a plasmid encoding firefly luciferase [111]. The hydrogel-containing living attenuated bacteria migrated and colonized to anaerobic tumor sites, effectively inhibiting the growth of tumors, including CT-26, B16, and large VX2, via activating the photosensitizer Chlorin e6 (Ce6). Furthermore, living engineered bacteria stimulated the strong immunogenicity as to promote the transition of anti-inflammatory macrophages from M2 type to M1 type, and activated natural killer cells and upregulated the expression of various effector cells to enhance their antitumor effect in tumor regions. 

Deng et al. [112] showed transgenic living bacteria and used photothermal therapy to create a single biological therapeutic cancer treatment. Synthesized *E. coli*(p)/PDA/Ce6 was able to significantly suppress the growth of osteosarcoma in tumor-bearing mice compared to *E. coli* without genetic engineering, which even completely inhibited the proliferation and growth of solid tumors. Engineered bacteria expressing cytolysin (ClyA) were used for the targeted treatment of pancreatic cancer [113], which was able to accumulate significantly and specifically in anaerobic tumor tissues and stimulate tumor immunity with infiltration of immune cells such as neutrophils, macrophages, CD4^+^, CD8^+^, and T lymphocytes. Meanwhile, ClyA promoted the expression of pro-inflammatory cytokines IL-1β and TNF-α at tumor sites. Afkhami-Poostchi et al. [114] reported a genetically engineered *E. coli* DH5α expressing in combination with glycyrrhetinic acid, which could promote the conversion of glycyrrhizic acid to metabolically active glycyrrhetinic acid and had the ability to significantly promote tumor cell apoptosis. Genetically engineered probiotics *E. coli* MG 1655 expressing exogenous glucose dehydrogenase were designed to actively target to solid tumor regions and consume glucose nutrition [98]. This work found that the engineered microorganism could effectively deplete glucose and induce tumor cell autophagy and p53-mediated cell apoptosis both in vitro and in vivo.

#### 3.1.3. Biomimetic Cell-Surface Coating

Living bacterial therapy has emerged as one of the novel wishes for the more effective treatment of various solid tumors. However, its clinical translation is greatly limited due to its worries of biosafety and high clearance rate in vivo. Biomimetic cell-surface coating may be a successful method to improve biocompatibility and reduce the elimination of living bacteria by macrophages [32,38,115]. A novel approach was introduced to enhance antitumor targeting by employing a red blood cell membrane-coated “invisibility cloak” for the probiotic *E. coli* Nissle 1917 (EcN) [116]. The retention of engineering EcN wearing a camouflage coat was significantly improved compared with wild-type EcN in vivo. Engineering EcN realized the precise imaging of solid tumors for up to 12 days via expressing luciferase in EcN, and the efficiency of accurate and long-term imaging was further superior to small molecules or other nano-imaging agents in vivo. Additionally, red blood cell (RBC) membranes could be wrapped onto attenuated *Listeria monocytogenes* (*Lmo*) [117]. *Lmo*@RBC generated immunogenicity through activating CD8^+^ and CD4^+^ T cells and eliminated tumor cells via producing NADPH oxidase-mediated reactive oxygen species (ROS) while retaining the capability of living *Lmo* to colonize the anaerobic tumor microenvironment. Interestingly, *Lmo*@RBC further enhanced its antitumor efficiency via inducing Caspase-8 to activate GSDMC-mediated pyroptosis. Simultaneously, the internalization of living *Lmo*@RBC in CT-26 and 4T1 tumor cells was greatly enhanced and the trouble of biosafety caused by bacterial infection and inflammation were especially reduced due to wrapping *Lmo* with the cell cloaking. Similarly, nanoshells derived from tumor cells could be utilized to coat bacteria. Another study reported that carcinoma cell membrane fused to the surface of attenuated *Salmonella typhimurium* VNP20009 was synthesized [100]. The biomimetic systems improved the biosafety of living microorganisms and reduced the clearance of phagocytes. More importantly, compared with single bacteria-based tumor therapy, tumor-associated antigens of tumor cell membrane in the nanoplatform elicited systematic antitumor immune responses in several tumor-bearing mouse models under intravenous administration. Meanwhile, the combination of tumor cell membranes with bacteria and immune checkpoints showed a superior improvement in suppressing the melanoma growth of metastasis. Therefore, coating with RBC membranes or tumor cell membranes to modify living bacteria may become an alternative and potential strategy for cancer biotherapy in a safe way.

Not only does the cell membrane serve as a camouflage invisibility cloak, the positively charged natural compound chitosan could conjugate to negatively charged living bacteria through electrostatic interactions. Li and co-workers constructed the engineered EcN expressed of the green fluorescent reporter gene (GFP) by cl857, a thermosensitive mutant of the cI protein from bacteriophage, as a gene circuit [118]. This thermosensitive engineered EcN wrapped with chitosan improved colonization viability due to the reduction in the macrophage-mediated clearance. Engineering systems camouflaged with chitosan enhanced tumor immunogenicity and effectiveness of tumor targeting under thermal stimulation. *E. coli* Nissle 1917 was widely employed in recent studies for bacterial antineoplastic therapy. An effective switch for the surface capsular polysaccharide (CAP) of living *E. coli* was designed [99], which could help to protect *E. coli* from assault including the phagocytosis and elimination of immune system. The probiotic was improved with genetic engineering to become invisible and more effective at delivering specialized medications directly to the solid tumor field. A small molecule called isopropyl-D-thiogalactopyranoside activated the *KFiC* gene, which in turn caused the production of CAP. This action controlled the smart occasion of engineered bacteria cleared by the immune system in human blood and organs, enabling living engineered *E. coli* to reach the additional solid tumor sites. Simultaneously, this bioengineering strategy greatly enhanced biocompatibility due to engineering living bacteria that could be more easily cleaned by the rest of the organism. The facultative anaerobic probiotic *E. coli* Nissle 1917 was comprehensively assessed as a micro-intelligent robot for tumor-targeted imaging and therapy [119], which showed a tendency to actively aggregate toward tumor sites and have a significant tumorigenicity inhibitory effect.

### 3.2. Engineering Bacterial Components as Drug Carriers for Cancer Therapy

Bacterial components exploited as cancer treatment platforms mainly include bacterial OMVs, BGs, and BSPs. However, research in the field of oncology therapy is hindered by natural bacterial components, which may pose potential biosafety issues and limit therapeutic efficiency. Therefore, combining bacterial components with engineering strategies such as genetic engineering, surface modification, and drug loading may provide new insights to alleviate these concerns (Figure 6). 

Bacterial OMVs extracted from attenuated bacteria via engineering strategies may reduce endotoxicity and improve biosafety [17]. Therefore, bacterial OMVs are considered to be safer than living and attenuated bacteria [39] and have become extensively studied biomimetic nano-anticancer vectors due to their nanosize, phospholipid bilayer structure, and inherent characteristics of activating the natural immune system (Table 1) [42,49,120,121]. Many studies have reported that bacterial OMVs released by *msbB* mutants in *E. coli* are less toxic [39,105,106,107]. In addition to the *msbB* gene, mutations of *lpxL1* in *N. meningitides*, deletion of *htrB* in *Shigella*, and mutants of *PA-m14* in *P. aeruginosa* can reduce the virulence of bacterial OMVs, and removal of the *msbA/Imp* gene and others can also obtain bacterial OMVs with reduced virulence [53,56]. 

Kim et al. first employed OMVs to treat cancer through immunotherapy. Genetic engineering-attenuated bacterial OMVs [39] derived from *E. coli* (∆*msbB*) were specifically targeted and accumulated in the tumor field while inducing the expression levels of antitumor cytokines CXCL10 and INF-γ eventually led to tumor regression. This novel therapeutic brought out a long-time excellent immune response to resist the second challenge tumor in primary challenge-cured mouse models with no obvious side effects. However, bacterial OMVs derived from attenuated microorganisms may reduce immune overactivation, such as toll-like receptor 4 (TLR4) mediated signaling pathways, which is the main reason of inflammation induced by bacterial OMVs [23,60,122]. 

LPS attached to bacterial OMVs might interact with TLR4 on the surface of DC cells, resulting in a potential obstruction of bacterial OMV uptake [60]. Therefore, tumor vaccines based on OMVs were modified through genetic engineering, which could specifically conjugate the DEC205 antibody, facilitate the recognition and uptake of DC cells, and stimulate the activation of cytotoxic T lymphocytes and memory T cells. Afterwards, engineered OMVs effectively suppressed the tumor growth and metastasis in melanoma-bearing mice. A parallel study [68] demonstrated that the whole basic fibroblast growth factor-modified OMVs derived from *E. coli* DH5α could promote tumor angiogenesis. Meanwhile, autoantibodies alleviated B cell resistance along with the enhancement of antigen presentation, resulting in produced durable tumor immunotherapy. 

In addition to genetic engineering, it was proposed that stable LPS-free OMV (dOMV) could be formed by adding sodium deoxycholate to the preparation process [123]. Compared to naturally derived bacterial OMVs, dOMV has lower levels of endotoxin and no hemolysis, which may reduce the severity of the immune response and modulate the expression levels of immune pro-inflammatory cytokines such as tumor necrosis factor-α (TNF-α), interleukin-6 (IL-6), and interleukin-β (IL-β). Polyoxyethylene 10 oil-based ether (Brij-96), another chemomodulator, showed high adhesion to lipid A, which is a hydrophobic component of LPS, and could reduce endotoxins [122]. In addition, it was shown that surface modification by biomineralization and bio-coating can reduce endotoxin levels in vesicles, unlike the approach of genetic engineering and chemical reagents [23,124].

A recent study [124] found that OMVs extracted from *E. coli* BL21 had a remarkable suppression on tumor growth. Soon afterwards, the bacterial OMVs were incubated with CaCl_2_ in order to reduce the excessive systemic immune stimulation by LPS. Biomineralized OMVs could neutralize and reprogram the acidic tumor microenvironment, greatly reducing the severe inflammatory reaction compared with natural OMVs. Simultaneously, biomineralized OMVs were highly effective at promoting macrophage M2-M1 polarization. The expression levels of immune cytokines were upregulated in tumor regions after biomineralized OMV intravenous injection. In this study, folic acid and indocyanine green combined with biomineralized OMVs were constructed to develop multifunctional tumor immune platforms for more efficient application in cancer treatment. Similarly, full advantages of the PEG/Se layer were taken to coat OMVs fused with CD47 nanobody [23]. The PEG/Se layer was disrupted when modified OMVs were delivered in solid tumor under radiation-triggered control, and the CD47 nanobody exerted the function to block CD47-mediated inhibition of efficient tumor cell phagocytosis by macrophages. The systematic overactivation of immune response was mitigated compared to the injection of natural OMVs. Meanwhile, the dose of modified OMVs was increased via intravenous administration due to the improvement of biosafety. PEG/Ser-coated OMV-CD47nb exhibited the ability to remodel the tumor microenvironment, including the promotion of M1 polarization and the change in protumor M2 polarization to antitumor M1 polarization; and the activation of long-time adaptive immune response, resulting in tumor regression and an efficient strategy to prevent tumor rechallenge. In addition to genetic/chemical engineering and surface modification of bacterial OMVs, artificial synthetic bacterial vesicles (SyBV) [40] were constructed to be greater immune adjuvants than classical excipients for cancer immunotherapy, with mild immune side effects compared to natural bacterial OMVs. *E. coli* SyBV combined with melanoma extracellular vesicles could activate dendritic cells, induce human tumor antigen-specific immune responses, and thus lead to tumor regression in melanoma tumor-bearing mice.

Chemotherapy drugs such as doxorubicin (DOX) combined with bacterial OMVs could result in superior antitumor efficacy. Gao et al. [22] proposed *E. coli*-derived bilayer membrane vesicles (DMVs) expressing Arg-Gly-Asp (RGD) and endogenous bacterial-targeting ligands. Simultaneously, DOX was loaded to enhance multiple tumor therapies through the pH gradient-driven method. DOX-loaded RGD-DMVs could actively target neutrophils and monocytes across the tumor vascular barrier in TME. DOX-RGD-DMVs significantly inhibited tumor growth compared with other groups in the B16F10 melanoma-bearing mice. In another research, doxorubicin-loaded OMVs (DOX-OMVs) were constructed from attenuated *Klebsiella pneumonia* ACCC 60095, and DOX was encapsulated in inner membrane of OMVs [102]. DOX-OMVs evoked intensive cytotoxic efficacy and caspase-dependent apoptosis in vitro, and substantially suppressed the growth of tumors in A549 xenograft BALB/c mice. Simultaneously, DOX-OMVs recruited macrophages and enhanced cellular uptake in TME. Sagnella et al. [125] developed nonocells, a type of bacterial OMVs [53]-loaded metabolite of chemotherapeutic drugs. These nanocells could activate NK cells, promote M1-type polarization of macrophages, and increase the expression levels of pro-inflammatory cytokines TNF-α, IL-6, and IL-12p40. These nanosystems induced the maturation and antigen presentation of BMDC cells, activated effector T cell immune responses, and thus generated powerful antitumor immune responses in tumor-bearing mouse models and clinical patients with malignant tumors. In another study, bacterial OMVs extracted from *Salmonella typhimurium* could effectively inhibit the proliferation of HTC116, MCF-7, and HepG2 in vitro [101]. Meanwhile, the treatment of bacterial OMVs combined with paclitaxel had significant antitumor ability in vivo, which increased NK cell infiltration and apoptosis of tumor cells. More interestingly, a study on a macrophage-mediated therapeutic system for delivering *E. coli* DH5α OMVs that co-loaded the photosensitizer Ce6 and DOX (DOX/Ce6-OMV@M) was assembled [126]. Bacterial OMVs were recognized and phagocytized by macrophages and neutrophils due to the existence of PAMPs [23,127,128]. This study demonstrated a synergistic chemotherapy/photodynamic therapy/immunotherapy co-delivery platform with few side effects, which upregulated the expression of pro-inflammatory cytokines such as TNF-α, IL-6, and INF-γ in 4T1 tumor-bearing mice and promoted the M1 polarization of macrophages, effectively promoting tumor cell apoptosis and inhibiting tumor metastasis. Similarly, a recent study [129] proposed that neutrophil hitchhiking could be utilized to deliver OMVs/DOX to tumor locations in glioma mouse models. The strategy significantly enhanced drug accumulation at the tumor sites. Additionally, OMVs derived from genetically engineered S.t-ΔpG^FlaB^ were able to silence P-gp protein expression and thus enhance the antitumor sensitivity of DOX, polarizing the M2-M1 phenotype of macrophages, leading to complete tumor eradication and 100% survival in tumor-bearing mice. The bacterial OMVs combined with chemotherapy have been employed in clinical trials and achieved decent experimental outcomes [125]. Therefore, bacterial OMVs could be used as an immune adjuvant to enhance the antitumor effect and improve limitations caused by chemotherapy drugs.

Bacterial OMVs could combine with photothermal therapy for cancer treatment. In a study [105], melanin-encapsulated bacterial OMVs possessed high photothermal conversion efficiency, which is very suitable for photothermal therapy. When melanin-OMVs were directly injected into tumor sites, the result exhibited the potential to inhibit tumor growth and could be applied for tumor photoacoustic imaging in view of the existence of melanin. The expressions of TNF-α, IL-6, and IFN-γ were significantly upregulated within a few hours, then subsequently downregulated to baseline after 24 h in vivo. This revealed better biosafety and a more appropriate immune response in tumor-bearing mouse models. Another study revealed [43] that a low dose of bacterial OMVs containing LPS could induce the extravasation of red blood cells in the tumor field and result in tumor regression under photothermal therapy. Therefore, the combination of bacterial OMVs and photothermal treatment may stimulate a more powerful antitumor immune response.

It is useful to combine bacterial OMVs as antitumor immune adjuvants with tumor cell membranes for personalized immunotherapy. A novel hybrid functional membrane vesicle (mTOMV) was successfully engineered by employing ultrasonic fusion techniques to combine membranes sourced from *E. coli* and 4T1 cancer cells [78]. Such biomimetic nanovesicles could significantly enhance innate immunity and promote dendritic cell antigen presentation and cellular uptake. mTOMV remarkably upregulated the expression of antitumor immune cytokines and activated splenic lymphocytes due to the presence of bacterial OMVs, whereafter was exhibited a specific inhibitory efficiency on the growth of 4T1 tumor cells. Researchers proposed that the potential of mTOMV for personalized tumor immunotherapy might be attributed to the presence of neoantigens from autologous cells. In this study, mTOMV could effectively inhibit tumor growth and lung metastasis in 4T1 tumor-bearing mice and significantly enhance the expression levels of INF-γ. Therefore, bacterial OMVs could also be modified to achieve multifunctional treatment in addition to combining with tumor cell membranes to form hybrid membranes to enhance personalized immunotherapy. Those pioneering studies present a significant advancement in the development of innovative membrane-based platforms in cancer therapy.

Hence, in order to obtain more robust tumor regression, bacterial OMVs could combine chemotherapy, photodynamic therapy, and personalized immunotherapy. The high systemic toxicity and systemic pro-inflammatory response of OMVs are generally attributed to LPS [53,56,130], whereas the antitumor immune response induced by bacterial OMV-based therapy was eliminated in IFN-γ-deficient transgenic mice [39]. Therefore, the presence of LPS on OMVs might be a double-edged sword, and it could also be a valuable tool for triggering antitumor immune response to suppress tumor growth and metastasis.

In addition to bacterial OMVs, BGs hold potential as drug carriers for targeted tumor vaccine delivery. Youssof et al. [131] employed BGs derived from *E. coli* BL21 to effectively deliver the chemotherapeutic agent 5-FU for colorectal cancer treatment. The BGs exhibited slow-release drug behavior and induced higher apoptosis in Caco-2 cells compared to free 5-FU. Additionally, Michalek et al. [103] demonstrated that BGs derived from *E. coli* Nissle 1917, due to an intact pathogen-associated molecular pattern, were capable of promoting DC maturation. Furthermore, BGs loaded with tumor lysates significantly increased CD4+ and CD8+ T cell proliferation and IL-12p70 cytokine expression, leading to the efficient recognition of tumor-associated antigens. This highlights the promising role of BGs as a vaccine tool for tumor immunotherapy. Another study [132] showcased BGs’ potential as a tumor vaccine platform, effectively stimulating DC cell maturation and displaying superior activation of CD8+ T cells compared to LPS and IFN-γ-induced groups. Notably, BGs demonstrated promising natural adjuvants in tumor immunity. Groza et al. [104] illustrated their synergistic antitumor activity when combined with oxaliplatin, resulting in prolonged survival and even complete remission in tumor-transplanted mice. This tumor adjuvant effectively activated a robust T-cell response and induced immunogenic cell death. The versatile characteristics of BGs in targeted drug delivery and their ability to enhance immune responses make them a valuable tool in tumor vaccine development and tumor immunotherapy. 

Bacterial components used as antitumor nanocarriers also include spores produced by bacteria or fungi. A study revealed that Ganoderma lucidum spores (GLS) exhibited inhibitory effects on the proliferation of human gastric cancer (SGC-7901), lung cancer (A549), and lymphoma (Ramos) cell lines. Notably, the observed inhibition is attributed to the presence of polysaccharides and triterpenoids within GLSs [133]. Meanwhile, the tumor cells labeled with CM-Dil were transplanted by the microinjection at the yolk sac site of zebrafish embryos, which has a similar antitumor effect in the tumor-bearing zebrafish embryo model. *Clostridium novyi-NT*, a Gram-positive obligate anaerobic attenuated strain without pathogenic alpha toxins, has been proven to have the potential to colonize solid tumor regions and suppress tumor growth [134]. A phase Ⅰ clinical trial showed that *C. novyi-NT* spore injection would activate the immune response and result in tumor regression in some of patients [41]. BSPs could combine other ingredients to produce more superb antitumor effects as nanocarriers for cancer therapy. In a research study, β-cyclodextrin and adamantine was wrapped on the surface of *C. butyricum* through host–guest interactions, which actively targeted and enriched the tumor sites of CT-26 tumor-bearing mice [93]. The presence of surface dextran contributed to the production of short-chain fatty acids, resulting in up to 89% tumor inhibition compared with simple treatment of BSPs, while oral administration instead of intravenous injection was safer and equally effective in this study. In addition, researchers also found that the effective tumor suppression was greatly reduced when antibiotics were used to disrupt the intestinal microenvironment. Tumor therapy of spores-dex was fundamentally based on the effective regulation of the intestinal microbial environment.

However, simple treatment of OMVs or other bacterial components could only produce tumor immunosuppressive results. The tough challenges are how to improve the efficiency of tumor treatment and long-term systemic toxicity, as well as enhance biocompatibility in vivo, which are still significant influencing factors for the further clinical translation of bacterial immunotherapy. Therefore, engineering strategies and nanotechnology combined with bacteria and bacterial components could be an unparalleled strategy for addressing the side efficacy of bacteria-based cancer therapy alone.

### 3.3. Bacteria and Bacterial Components as Nanocarriers for Gene Delivery in Cancer Therapy

Due to the invasion of coronavirus disease-19 (COVID-19), the development of gene drug treatment becomes extraordinary rapid. Additionally, the use of vaccines as one of the effective measures prevents COVID-19. Gene therapy is rapidly becoming a prominent field in cancer treatment, with a growing number of related studies being gradually reported [135,136]. For example, liposomes have been successfully explored to deliver mRNA approved by the FDA-BNT162b2 (BioNTech/Pfizer) and mRNA-1273 (Moderna) [137]. Nevertheless, liposomes with mature preparation technology and superior biocompatibility have different challenges for clinical application, whereby they do not have therapeutic effects and complex surface modification. Bacterial components such as bacterial OMVs are equipped with similar structures compared to liposomes and have distinctive functions of immune adjuvants [17,138,139]. Hence, combining gene therapy with bacterial therapy may create more remarkable antitumor-targeting effects (Table 2). Bacterial OMVs as biomimetic nanocarriers may provide an insight platform for the co-delivery of nucleic acid genetic medicines and chemotherapeutic drugs. OMVs released from *E. coli* BL21 (Δ*msbB*) were employed to load DNA damage response 1 (*Redd1*)-siRNA via electroporation, and could be modified with paclitaxel on the surface [106]. This system could first release paclitaxel in the tumor field and promote tumor-associated macrophage repolarization, then aggregate the activated mighty tumor immune response through promoting activation of T lymphocytes and promoting the maturation of dendritic cells. Meanwhile, the nanosystem upregulated the expression levels of anticancer immune cytokines, including IFN-γ, TNF-α, and IL-12, and downregulated the expression of IL-10. Therefore, the synergetic nanoplatform possessed a satisfying antitumor capacity that almost completely suppressed the growth and metastasis of tumors and had no significant systematic toxicity in 4T1 tumor-bearing mice. Generally, lipid NPs are used for mRNA vaccine delivery by microfluidics, for instance, the SARS-CoV-2 and COVID-19 vaccine as drug delivery carriers [140,141,142] with gene medicines for clinical transformation, while some reports using “Plug-and-Play” technology are expected to further promote the development of nucleic acid drug treatment. For example, engineering OMVs (OMV-LL) produced by *E. coli* BL21 via genetic engineering and molecular glue technology were manufactured [59] that could strongly stimulate the immune system, promoting antigen presentation and activating T cells due to the abundant PAMPs onto OMVs [122]. Meanwhile, OMV-LL could conjugate L7Ae-bound box C/D-mRNA antigens for delivery into DC cells for cross-presentation via Listeriolysin O-mediated endosomal escape. In this study, outcomes have shown that OMV-LL-mRNA could significantly inhibit the growth of melanoma tumor-bearing C57/BL6 mice, and brought 37.5% of colorectal tumor-bearing C57/BL6 mice to complete recovery. 

In a recent study [144], bacterial OMVs were extracted and purified by engineered *E. coli* K-12 W3110 modified with LyP1; the programmed death-1 (PD-1) plasmid was subsequently transfected into this drug vector. After genetic modification, the Myc-tag protein was produced, which decreased the systemic toxicity brought on by the LPS. The recruitment of cytotoxic T lymphocytes (CTLs) and natural killer cells as a result of inhibiting the immunological checkpoint PD-1 and its main ligand PD-L1 binding to bacterial exosome proteins causes the release of IFN-γ cytokines in tumor tissue. Finally, LOMV@PD-1 with remarkable biocompatibility showed a more excellent inhibition rate of 4T1, CT-26, and B16 tumors in vitro and in vivo compared with other groups such as pure PD-1 antibody and OMV@PD-1 without genetic engineering modification. CD38, an NAD glycohydrolase, is closely associated with the early and late stages of tumorigenesis. In another work, PLOVs were designed by fusing attenuated *Salmonella* OMVs and photosensitive liposomes (PLOV) [120]. Then, CD38 siRNA was loaded to form CD38 siRNA@PLOV via ultrasound, which tends to target T cells and then downregulate CD38 expression at the tumor sites. Moreover, PLOV significantly promoted the maturation of BMDC cells due to the presence of OMVs. The expression levels of TNF-α, IL-6, and IL12p70 were upregulated in vivo and in vitro, and the infiltration of immune CD4 and CD8 T cells was detected in tumor tissues under laser irradiation. Additionally, the combination of anti-PD-1 effectively inhibited tumor growth and metastasis in distant and primary H22, CT-26, and 4T1 tumor-transplanted mice. Meanwhile, CD38siRNA@PLOV-based photothermal therapy plus anti-PD-1 stimulated the tumor-bearing mice to produce a tumor vaccine effect. This study provided a new research basis for the powerful tumor immunity of bionic small nucleic acid medicines based on the hybrid membrane of bacterial OMVs and photosensitive liposomes. 

Bacterial minicells are types of bacterial OMVs with 100~400 nm size, which do not contain the chromosome compared with the parent bacteria [53]. For example, EGFR-targeted bacterial minicells were synthesized to deliver siRNAs (si*PLK1*, si*KSP*, si*CDK1*, and si*MDR1)* via simple incubation owing to specific protein channels in vitro and in vivo [145]. The presence of minicells was effective in enhancing the innate immune response of the organism, increasing the expression levels of TNF-α, IL-6, and INF-α, β, γ. EGFR-modified bacterial minicells loaded with siRNA could validly knock down the expression of *PLK1*, *KSP*, *CDK1*, and have the excellent ability to suppress tumor cell proliferation in HCT116 tumor-bearing mice. Meanwhile, EGFR minicells loaded with si*MDR1* reversed drug resistance in Caco-2/*MDR1* tumor-bearing mice and MDA-MB-468/*MDR1* tumor-bearing mice, leading to marked tumor inhibition, and improve the sensitive ability of these tumors with the *MDR1* gene to chemotherapeutic drugs. 

In addition to bacterial components, bacteria could be employed as tumor-targeting nanocarriers to deliver gene therapeutic medicines to solid tumor tissues. Recently, a novel bacterial nanomaterial capable of targeting the tumor-specific biofilm was developed to regulate the TME [143]. The nanomaterials consisted of a tetrahedral framework nucleic acid chemically linked the nucleic aptamer AS1411 and 5-fluorouracil, which could effectively aggregate and permeate into the deeper sites of the tumor. In addition, *Streptococcus mutans* was inactivated and used as a biological carrier to cooperate with the targeting of tumor-specific biofilm and stimulate the maturation of DC cells and promote the T-cell antitumor immune response, synergistically resulting in inhibiting the progression of the tumor. Briefly, engineered bacteria and bacterial components loaded with gene medicines may be an innovative strategy for the further treatment of solid malignancies.

## 4. Engineering Strategies for Combination of Nanotechnology and Bacteria-Based Drug Systems for Cancer Treatment

Nanotechnology cooperating with engineered bacteria possesses more of a promoting multifunction, enabling prospective biomimetic nanocarriers to migrate to the deeper tumor tissues that traditional medicines cannot reach and achieve a more therapeutic effect to suppress tumor progression and metastasis [29,32,146]. Living bacteria tend to migrate, grow, and colonize toward the hypoxic and low-pH TME, whereas strong immunogenicity may lead to biosafety issues. Probiotics are commonly utilized as sources of engineered bacteria, including *E. coli* Nissle 1917, *Saccharomyces boulardii*, *Lactobacillus reuteri*, *Lactobacillus casei*, *Lactobacillus rhamnosus*, *Bifidobacterium infantis*, and *Bifidobacterium breve* [37]. However, the clinical transformation of living bacteria turns greatly limited due to the lack of precise strategy to control the smart release of drugs and the capability to maintain adequate doses for remotely inhibiting tumor growth after administration. Therefore, living bacteria combined with nanotechnology could construct engineering microorganisms as therapeutic biomimetic drug nanocarriers. Simultaneously, bacteria-NPs cooperating with traditional chemotherapeutics may achieve superior therapeutic efficacy [29,146]. The combination of bacteria and NPs mainly consists of four methods: chemical bonds, electrostatic interactions, biotin–streptavidin, and other binding forms (Table 3 and Figure 7). 

### 4.1. Chemical Bonds

The structure of peptidoglycan on the bacterial cell wall provided reaction sites of chemical grafting for conjugating NPs to form a hybrid platform [154]. Amide-bond and polydopamine (PDA) coupling are the most common chemical bonds for conjugating NPs onto the surface of bacteria [96]. A research study [155] proposed that Bi_2_S_3_ nanoparticles (BNPs) containing amino acid were chemically modified by an amide-bond condensation reaction onto the surface of *E. coli* MG1655 overexpressing ClyA protein. This bacteria-NP system (Bac@BNP) equipped with viable and hypoxic tropism could autonomously move target to 4T1 tumor sites, secreted ClyA protein to regulate the tumor cell cycles to G2/M and G0/G1, and amplified the antitumor effect under X-ray irradiation. Therefore, Bac@BNP triggered 4T1 cell reactive oxygen species (ROS) generation and resulted in DNA damage. Compared with other groups, Bac@BNP plus X-ray significantly inhibited tumor growth and metastasis in 4T1 tumor-bearing mouse models. Bac@BNPs boosted antitumor immunity, resulting in the maturation of DC cells and T-cell activation. This work provided an innovational strategy for intelligent living bacteria combined with NPs to improve antitumor radio-sensitization. A recent study [149] put forward hybrid nanobacteria through self-assembly cisplatin prodrug nanocapsules conjugating with *E. coli* MG1655 via amination reaction. *E. coli*-loading nanocapsules moved and colonized to the tumor sites and exhibited significant anticancer ability in vitro and in vivo. The bacterial hybrid nanocapsules enhanced T-cell infiltration and pro-inflammatory cytokine expression levels and showed excellent biocompatibility under NIR irradiation, because the living bacteria could be killed by the high temperature conducted by photothermal therapy after the bio-hybrid nanoplatform selectively enriched in solid tumor sites. 

In the another recent study [147], the engineered bacterial microrobots realized magnetothermal and multifunctional therapeutic effects in a CT-26 tumor bearing mice model via chemical amide bonds and specific genetic circuit construction. The engineered probiotic *E. coli* Nissle 1917 carried magnetic nanoparticles (EcN@MNP), and a thermal logic circuit was used as a temperature and localization reporter, and NDH-2 enzyme was encoded in EcN for enhanced antitumor therapy. The bacteria-NPs showed good thermosensitivity and actively targeted the tumor region, and promoted CT-26 tumor apoptosis both in vitro and in vivo under magnetothermal therapy. Similarly, Ma et al. [156] reported a micro–nano biorobot based on bacteria, which holds promise for tumor diagnosis and treatment. The researchers utilized Fe_3_O_4_@lipid nanocomposites to modify the engineered bacteria through amino groups, which resulted in more drug accumulation at orthotopic colon tumors site in tumor-bearing mice. By converting magnetic signals into heat via the use of paramagnetic Fe_3_O_4_ NPs, the bacteria were activated to express lysis proteins, which were under the control of a heat-sensitive promoter. The modified bacteria were then lysed, releasing their internal, pre-expressed anti-CD47 nanobody payload, producing a potent antitumor response in female mice against both orthotopic colon tumors and distal tumors. The magnetically modified bacteria also facilitated a continual magnetic field-controlled motion for the improvement of tumor-targeting ability and higher therapeutic efficacy. These findings represent an innovative approach in the field of biorobotic magnetothermal therapy for cancer treatment, and may have significant implications for future therapeutic interventions. Another interesting study [157] synthesized liposome-encapsulating lysovirus (OAs) through biocondensation reaction, attaching active ester groups to the surface of *E. coli* BL21 with amine groups. *E. coli*-lipo-OAs actively delivered OA to the solid tumor region through the tumor-homing action of the living bacteria. The bacterial–viral couples enhanced T cell infiltration and the recruitment and maturity of DC cells in primary and distant tumors via the activation of tumor immunogenicity. 

Consistent with the above findings, tumor-specific ovalbumin antigen (OVA) and checkpoint-blocking antibody α-PD-1 with dopamine were assembled by in situ co-deposition polymerization to form a complex, denoted as PDA [158], which subsequently coupled to the surface of bacteria. EcN with multiple immune-activating characteristics activated the innate immune response and converted the anti-inflammatory phenotype M2 of tumor-associated macrophage cells to the antitumor phenotype M1. Meanwhile, OVA promoted DC cell maturation and activation of antigen-specific T-cell responses. Afterwards, EcN tended to target anaerobic tumor sites and significantly inhibited tumor cell growth, leading to tumor regression and increased survival time in two tumor models (CT-26-OVA and MC38-OVA). This study provided a novel research strategy for multiple combination immunotherapy through bacteria combined with nanotechnology. The current study introduces a pioneering methodology for combating cancer and viruses concurrently, through the use of tumor-resident bacteria that are adorned with a hybrid immunoactive nanosurface [159]. The method involves the co-deposition of dopamine to generate PDA nanoparticular linkers, which are then combined with immunoactive PD-1 antibody and antigenic S1 protein to produce the nanosurface. This unique linker can interact with amino acid residues on the bacterial surface and immunoactivators without the need for a catalyst. The clothed bacteria can effectively infiltrate and colonize the tumor tissue, releasing and amplifying PD-1 and S1 protein to activate immune cells. The suggested approach results in a dual immune response that is antiviral and anticancer, making it a promising approach in the treatment of viral infections and malignancies. Another study [160] described the development of a microbial smart nanorobot, namely LOD/TPZ@Lips-LA, which exhibited a remarkable capability to migrate and thrive in the hypoxic regions of tumors. This nanorobot was fabricated by conjugating liposomes containing lactate oxidase (LOD) and the chemotherapeutic drug tirapamil (TPZ) to the surface of Lactobacillus (LA) via an amide reaction. Upon reaching the anaerobic tumor environment, LOD/TPZ@Lips-LA was triggered to release its cargo, which in turn effectively suppressed tumor growth while minimizing damage to healthy tissues. Calreticulin, heat shock protein 70, and High-mobility group box expressions were upregulated due to the administration of microbial smart nanorobots, which accelerated immunogenic cell death. Following that, the ICD status promoted T-cell infiltration and macrophage M1 type polarization at the tumor sites, resulting in synergistic antitumor effects of chemotherapy and immunotherapy. A recent report demonstrated that Trojan bacteria were able to effectively bypass the blood–brain barrier [161] for delivering targeted drugs to the brain. Glucose polymer silica NPs were loaded with the photosensitizer indocyanine green attached to the surface of attenuated *Salmonella typhimurium* (VNP20009) and *E. coli* (ATCC 25922) via Schiff base reaction. Trojan bacteria triggered photothermic effects and thus significantly inhibited the growth of glioblastoma G422 cells under NIR laser irradiation. Afterwards, Trojan bacteria combined with photothermic therapy promoted DC cell maturation and significantly increased the proportion of CD8^+^ T cells and NK T cells and the expression levels of TNF-α and IFN-γ. In all, the Trojan system induced both innate and adaptive tumor immunity, leading to penetrating the blood–brain barrier and producing the potent effectiveness of tumor suppression in the solid tumor region. Furthermore, the residual Trojan bacteria were effectively eradicated by the immune system, thus demonstrating outstanding in vivo biosafety.

### 4.2. Electrostatic Interactions

Electrostatic interactions could occur between negatively charged bacteria and positively charged NPs [27,96]. Cationic polymers such as chitosan and polyetherimide are used to encapsulate both water-soluble and poorly soluble drugs, which can greatly enhance their stability and drug-loading ability [162,163]. *Salmonella typhimurium* (VNP20009), a facultative anaerobic bacterium, could proactively direct and colonize to tumor sites, and thus serve as a functional drug delivery system [164]. Attenuated *Salmonella* coated with antigen-adsorbing polyamidoamine dendrimer nanoparticles via electrostatic interactions was developed [150], which elicited a systemic antitumor immune response that suppressed the growth of CT-26 tumors after administration. The anticancer mechanism may be explained by the fact that living bacteria could transport tumor antigens to promote the maturation of DC cells around the tumor regions. Engineering *Salmonella* improved biosafety through deletion of the *msbB* gene. This study confirmed that bacteria-NPs could enhance immune responses by means of maturation of DC cells and activation of T cells. Meanwhile, engineered attenuated bacteria could reduce their toxicity and retain their powerful immune stimulatory properties, which has become an effective vaccine for potential cancer therapy [165]. 

Similarly, bacteria could activate the immunogenicity of the body and increase the expression of inflammatory cytokines along with the maturation of DC cells. Chitosan-coated genetically engineered *E. coli* DH5α bionic microcapsules were designed to carry protein drugs [166]. Positively charged chitosan was wrapped around the surface of negatively charged *E. coli* via electrostatic interactions. In addition to sustaining regulated release for two weeks, chitosan *E. coli* microcapsules were able to stimulate specific immunity to B16-OVA tumor cells. While this system was able to activate CD8^+^ T cells and promote the maturation of DC cells, it upregulated the expression of pro-inflammatory cytokines TNF-α and IL-12P40 and resulted in inhibiting tumor growth in B16-bearing mice. The combination of hybrid systems of bacteria and NPs could combine with photodynamic therapy to achieve synergistic antitumor efficacy. Single nanomedicines are limited in cancer treatment due to the insufficient drug uptake and tumor targeting in the dense solid tumor sites compared to bacteria-based nanoplatforms. 

*E. coli* Nissle 1917 is widely used to bind with nanoparticles for deeper tumor uptake and multifunctional antitumor effects. One study [167] first reported that *E. coli* Nissle 1917 expressed catalase linked to black phosphorus quantum dots (BPQDs) via electrostatic adsorption to obtain the goal of effective therapeutic treatment in solid tumor regions. *E. coli*/BPQDs actively transferred into anaerobic tumor tissues in CT-26 tumor-bearing mice through intravenous injection. Afterwards, BPQDs stimulated reactive oxygen species to generate the excellent function of killing CT-26 tumor cells in vivo under laser irradiation. An engineered biohybrid nanomaterial containing paclitaxel and BAY-876-conjugated human serum albumin nanodrug (HPB) was integrated with *E. coli* Nissle 1917 via electrostatic interactions (EcN@HPB) [168]. Due to bacterial respiration, EcN@HPB might aggressively target CT-26 tumor locations and competitively deplete glucose, while BAY-876 further hinders the uptake of glucose by tumor cells through blocking the glucose transporter protein 1. Additionally, the internalization of HPB in tumor cells was markedly facilitated by the presence of HSA. Ultimately, the biohybrid nanoplatform improved the antitumor effects of chemotherapy via increasing tumor cell internalization. The conjugation of engineered bacteria with multifunctional nanoparticles may affect drug delivery efficiency due to excessive particle size. Therefore, a bio-targeting synergistic nanoplatform was constructed [169] using genetically modified *E. coli* that carried acoustic reporter genes that could encode gas vesicles. Engineered *E. coli* actively targeted and colonized solid tumor sites and produced ultrasound imaging capabilities as well as ultrasound ablation for tumor therapeutic efficacy. Multifunctional cationic LNPs containing several components were self-assembled by electrostatic conjugation with genetically engineered bacteria in the tumor region to achieve targeted multimodal imaging and improved the efficacy of synergistic tumor cell killing in 4T1 tumor transplant mouse models. This innovative concept could offer a new therapeutic avenue and promising clinical diagnosis for tumor ultrasound and bacterial therapy. In a separate study [170], a smart bacterial bioreactor containing NPs loaded with DOX linked with nonpathogenic *Shewanella oneidensis* MR-1 via electrostatic interactions was designed to carry the NPs actively targeted and aggregated at the site of an oxygen-depleted solid tumor, metabolizing lactic acid via bacterial respiration. Depletion of lactate within the tumor facilitated downregulating the expression of multidrug resistance-associated ABCB1 protein (also known as P-glyco protein in tumor cells) and prevented DOX efflux from tumor cells. This collaborative strategy contributed to the efficiency of chemotherapy and lactate metabolism therapy in the synergistic battle against oncology.

Yang et al. [171] developed 2D CoCuMo layered-double-hydroxide (LDH) nanosheets coated on *Lactobacillus acidophilus* (LA) probiotics via electrostatic interactions to build a TME-responsive platform for precision near-infrared (NIR) photodynamic treatment. CoCuMo-LDH nanosheets’ photodynamic activity for singlet oxygen generation under 1270 nm laser irradiation can be improved by TME-induced in situ amorphization. Tests conducted showed that LA and LDH can completely trigger cell death and eradicate tumors when exposed to 1270 nm laser radiation in vitro and in vivo. According to the study, probiotics can be used as a platform for highly efficient, precise NIR-II PDT tumor targeting. These results showed that NPs coated onto bacteria surface via electrostatic interactions have considerable promise for the development of potent cancer treatments.

### 4.3. Biotin–Streptavidin

Binding forms encompass bioaffinity or specific binding, such as biotin–streptavidin and antigen–antibody binding [21,27]. In one study [148], Akolpoglu and colleagues reported that magnetic materials and radionuclides (mNPs) with streptavidin were coated on the bacterial surface of *E. coli* MG1655 expressed GFP and biotin followed by biotin–streptavidin. Afterwards, the double complex was combined with liposomal NLs loaded with DOX and a photothermal agent to form a triple biotin–streptavidin–biotin complex. It was found that the ability of the bacterial biohybrid to swim through the HT-29 3D tumor spheres, using type I collagen to construct, was significantly improved upon the magnetic field. Meanwhile, the magnetic triple biotin–streptavidin–biotin complex could activate the release of DOX and enhance the death of spherical cancer cells through the photothermal effect under near-infrared irradiation. Another study [172] constructed *Salmonella typhimurium* carrying paclitaxel-encapsulated liposome via biotin–streptavidin. These engineered bacteria-based microrobots produced better suppression of 4T1 tumor cell growth in vitro, and the bacteria-loaded liposomes exhibited higher mobility and better biocompatibility than paclitaxel-encapsulated liposomes alone. Similarly, the engineered *Salmonella* attached to temperature-sensitive liposome contained DOX was synthesized [151], and could actively move to colorectal tumor regions and trigger the release of DOX. Meanwhile, it promoted M1-type polarization of macrophages and upregulated expression levels of pro-inflammatory cytokines TNF-α, IL-1β, and IL-10 under ultrasound conditions, as well as enhancing immune cell infiltration. These bacteria nanoparticles in turn induced excellent tumor-suppressive effects in vivo.

### 4.4. Other Binding Forms

In addition to chemical bonds, electrostatic interactions, and biotin–streptavidin, there are several studies combining bacteria and NPs with simple physical interaction and electroporation. Recently, a multifunctional biomimetic drug vector consisting of genetically engineered *E. coli* MG1655 and black phosphorus (BP) NPs was designed [173]. The *E. coli* was genetically programmed with therapeutic tumor necrosis factor-associated apoptosis-inducing ligand, allowing the therapeutic protein to be delivered directly to the tumor sites and thus induce tumor cell apoptosis. In addition, the researchers combined BP with this bacterial vehicle through simple physical interaction. Under laser irradiation, the bacteria could receive photoelectrons generated by BP NPs on their surface, which in turn released NO precisely at the tumor sites, enhancing the therapeutic effect and promoting the polarization of tumor-associated macrophages towards an antitumor M1 phenotype. Simultaneously, the production of reactive oxygen species induced immunogenic cell death, further enhancing antitumor efficacy. Furthermore, the biological system improved immunological effects via promoting tumor cell apoptosis, activating T-lymphocytes and releasing pro-inflammatory cytokines, which provided the basis for multifunctional antitumor bacterial biotherapy. 

Similarly, Reghu et al. [174] described a living attenuated *Bifidobacterium bifidum* (BB)-based inventive study via the incubation and washing method. The study developed Cremophor EL (CRE) for encapsulating organic dye molecules with BB. This study’s deployment of straightforward CRE coatings as bacteriotherapy poison reduction strategies rather than sophisticated genetic engineering represented a significant advancement when compared to previous studies. Genetic engineering may affect the proliferation and movement of engineering bacteria. As a result, the functionalized *Bifidobacterium* possessed superior fluorescence, high photothermal conversion efficiency, low toxicity, and excellent targeted antitumor capability. ICG-CRE-BB increased the expression of TNF-α and caspase-3, thus significantly inhibiting tumor growth in CT-26-bearing mice via stimulating immune responses upon near-infrared laser. Such reports show that the excellent potential and positive expectation of living bacterial antitumor therapy enables the complete curing of solid tumors. Electroporation could bring out the precise control of bacteria nanoparticles and modestly affect the viability of bacteria. For example, paclitaxel liposomes with weak negative charge could more easily enter the bacterial membrane into *E. coli* by electroporation, and the electric transfer efficiency was up to 95% [175]. *E. coli* loaded with paclitaxel liposomes could be internalized into A549 cells rapidly through endocytosis. The expression levels of VEGF, HIF-1α, and Bcl-2 were significantly downregulated and the expression levels of pro-inflammatory cytokines TNF-α, IL-4, and INF-γ were upregulated. Meanwhile, the efficient apoptosis of tumor cells was greatly improved. Whereafter, this nanoplatform showed significant tumor suppression in primary lung tumor-bearing mice after administration in mice.

## 5. Combination of Nanotechnology and Bacterial Component-Based Drug Delivery Systems

Nanotechnology cooperating with bacterial components such as bacterial OMVs, BGs, and BSPs possesses a potential multifunction, enabling these prospective biomimetic nanocarriers to arrive at the deeper tumor tissues that traditional medicines cannot reach and achieve innovative therapeutic effects to suppress tumor cell growth and metastasis. The advantages and challenges of nanotechnology of combination with bacterial components and nanoparticles as listed in Table 4.

### 5.1. Bacterial OMV Nanoparticle-Based Nanoplatforms

Bacterial OMVs have garnered significant attention as biomimetic components in anticancer nanocarrier research, owing to their unique extra structure derived from bacteria. Exploiting the immunogenic properties of bacterial OMVs, they have been harnessed as efficient immune adjuvants, synergistically combining with nanotechnology for potent anticancer therapy in conjunction with chemotherapy and photothermal therapy. Alongside bacterial OMVs, engineered cell membranes have emerged as promising platforms for drug delivery and cancer therapy applications. Erythrocytes, leukocytes, stem cells, cancer cells, platelets, endothelial cells, and other engineered membrane cells have demonstrated great potential in facilitating efficient drug delivery when integrated with NPs [179,180]. Three common methods are developed about cell membrane vesicles coated onto nanoparticles: membrane extrusion [152,181,182,183], ultrasonic fusion [184,185,186], and other binding forms (Figure 8 and Table 3) [187].

#### 5.1.1. Membrane Extrusion

The coating of bacterial OMVs onto the surfaces of NPs through membrane extrusion represents a widely employed technique in drug delivery research. This method remains one of the most extensively used strategies for integrating the unique properties of OMVs with NPs. Typically, the suspension comprising OMVs and NPs was forced through a nanosized polycarbonate membrane by co-extrusion vesicles seven times/twenty-two times [152]. The mini-extruder has become the most common employed equipment for the membrane fusion process, and the uncovered OMVs could be removed via high centrifugation [176]. The extrusion technique with various membranes of different sizes and qualities affects the particle size and polydispersity of the synthesis of nanomaterials [188]. For instance, a bioengineering method was employed to prepare OMV-DSPE-PEG-RGD-coated F127 tegafur-loaded nanomicelles (ORFT) [152]. OMVs generated by attenuated *Salmonella typhimurium* wrapped onto tegafur-based polymeric micelles by co-extruding vesicles and DSPE-PEG2000-RGD were conjugated on the surface of OMVs through chemical modification. ORFT combined with chemotherapy, bacterial immunotherapy, and nanotechnology could inhibit 70% tumor growth. ORFT activated macrophages to stimulate cytotoxic lymphoid T cells and improved the survival time of B16 tumor-bearing mice, effectively inhibiting tumor lung metastasis. At the same time, the immune-specific ORFT exercised the potential of vaccines in tumor-free mice, which could significantly prolong the tumor-free time after B16F10 melanoma transplantation. Bacterial OMVs as drug carriers could combine with immune checkpoint therapy, and have achieved excellent therapeutic effects. Meanwhile, the development of a nanosized polyplex coated with a bacterial membrane derived from *Mycobacterium smegmatis* resulted in nonpathogenicity and high immunogenicity. This innovative approach led to the production of a systemic antitumor immune response, facilitated by the enrichment of bacterial membrane constituents with abundant PAMPs [77]. Bacterial membrane-coated PC7A/CpG NPs combined with radiotherapy resulted in significant tumor regression in melanoma and glioma-engrafted mice. This enhanced the uptake and cross-presentation in DC cells and stimulated tumor T cell effects.

Poly (lactic-co-glycolic acid) (PLGA), an exemplary biodegradable polymer, has found extensive application in various long-acting drug formulations, which have been approved by FDA [189,190]. And PLGA could be employed as the core material to support bacterial OMVs. A recent study [123] proposed for the first time to extract LPS-free OMVs from *E. coli* K1 equipped with outer membrane proteinA (OmpA), which could bind to gp96 on BBB endothelial cells, therefore helping EC-K1 cross BBB and improve invasion ability. Meanwhile, OmpA could mediate the endosomal escape of NPs. OMV-NPs prolonged the blood clearance of the drug with superior BBB penetration and brain-targeting ability. This study provides a promising strategy for intracranial tumor treatment. To enhance the synergistic antitumor efficacy, several studies have integrated bacterial membranes or bacterial OMVs as immune adjuvants with tumor cell membranes enriched with specific tumor antigens [191,192]. Li et al. [183] reported the creation of adjuvants, named BTs, for cancer nanovaccines by leveraging membrane extrusion and ultrasonic fusion to fuse OMVs produced from *E. coli* with B16F10 cancer cell membranes. Three common polymers (PLGA, SiO_2_, and colloidal gold) were utilized as the core materials to support synthesized hybrid vesicles, which improved the stability of BTs. The bacterial membrane components in BTs@PLGA boosted DC cell maturation and antigen presentation, while the distinctive tumor endogenous antigens of the tumor cell membrane enabled BTs to produce significant and specific therapeutic effects in B16 F10 tumor-bearing mice. Meanwhile, the expression levels of pro-inflammatory cytokine IL-12P40 and IL-6 were upregulated. 

Biohybrid membrane nanoplatforms (MGTe) containing bacterial outer membrane (BM) extracted from *E. coli* MG1655 and 4T1 tumor cell membranes (TM) were designed [193], which bound to Glutathione (GSH)-modified Tellurium (Te) NPs by means of sonication and extrusion. BM and TM-induced T cell infiltration and immunogenic cell death brought on by X-ray sensitization. MGTe could activate the immune system, thus increasing the expression of the high-mobility group protein B1, adenosine triphosphate, and calreticulin. In contrast, hybrid nanoplatforms plus X-ray considerably evoked the DC maturation and upregulated the expression of representative immunostimulatory cytokines, including IL-6, IL-12, TNF-α, and INF-γ. Meanwhile, biohybrid systems significantly converted macrophages to the antitumor M1 polarization so as to modulate an immunosuppressive tumor microenvironment. Afterwards, MGTe-based X-ray sensitization further revealed the most splendid 4T1 tumor suppression in primary and distant tumor-bearing mice models owing to the great enhancement in antitumor immunogenicity. Another study [192] constructed similar nanoplatforms of hybrid membrane vaccines, which fused melanoma cytomembrane with bacterial OMVs derived from attenuated *Salmonella*. Since it is difficult to achieve tumor eradication with immunotherapy alone, the researchers utilized PLGA-ICG (PI) NPs as the core of the hybrid membrane to activate photothermal therapy. This system synergistically activated immunogenic cell death and promoted maturation of BMDC cells in B16-F10 tumor-bearing C57BL/6 mice under laser radiation. The maturation of BMDC cells was mainly attributed to the existence of PAMPs on the surface of bacterial OMVs. PAMPs were recognized by the pattern recognition receptors of DC cells, which enhanced the uptake of OMVs. Interestingly, the vaccine associated with B16-F10 tumor-specific antigens specifically and significantly inhibited tumor growth and metastasis in B16F10 tumor-bearing mice and greatly prolonged survival time, while there was no significant tumor suppression efficacy in 4T1 tumor-bearing mice. It was revealed that this hybrid membrane vaccine possessed a special antitumor ability due to the composition of tumor-specific antigens. Bacterial OMVs can activate an intrinsic immune response due to their exogenous “danger signal”. Chen and colleagues [191] combined an *E. coli* DH5α cell plasma membrane with a CT-26 tumor cell membrane via membrane extrusion to form a hybrid membrane vaccine, and used PLGA as the core of the support membrane. The hybrid membrane tumor nanovaccines (HM-NPs) were able to maximize the maturation of BMDC cells, and promoted the uptake of tumor antigens. Moreover, HM-PLGA NPs activated expression of TLR protein, which further activated the NF-κB signaling pathway and upregulated the expression levels of pro-inflammatory cytokines IL-6, TNF-α, and IL-1β. The final HM-PLGA NPs almost completely inhibited 4T1, CT-26, B16F10, and EMT6 tumor regression in tumor-transplanted mice. Consistent with the above study, HM-PLGA-NP vaccination was able to prevent tumor recurrence in CT-26 tumor-bearing mice because of the presence of the CT-26 tumor antigen. Therefore, the combination of tumor antigens and bacterial immune adjuvants as the components of the tumor vaccine could produce a powerful tumor suppression effect and reduce the possibility of a tumor recurrence effect in vivo. The strategy of fusing tumor membranes and bacterial membranes provided a new idea for the future development of tumor vaccines. 

Membrane extrusion is an effective method to wrap bacterial OMVs onto NPs. The synthetic nanomaterials are equipped with uniform size and tend to preserve biomolecules and proteins of bacterial OMVs well. Although membrane extrusion is commonly employed in the combination of biomimetic membranes and nanoparticles, the process is difficult and time-consuming, and the yield of the biomolecules is easily reduced, since mixtures must pass through the films through extrusion several times [176,194].

#### 5.1.2. Ultrasonic Fusion

In addition to membrane extrusion, ultrasonic fusion is also a commonly employed method in bacterial OMV fusion. Ultrasonic energy, the frequency of ultrasound, and ultrasonic time have significant effects on the characteristics of bacterial membrane nanoparticles such as size and distribution. Generally, a certain percentage of mixtures of OMVs are wrapped onto NPs through an ultrasonic probe or ultrasonic bath. The biomimetic nanoparticles might then be precipitated during centrifugation to eliminate any uncoated bacterial OMVs. For example, OMVs generated by *E. coli* DH5α and cancer cell membranes produced by B16-F10 cells were sonicated to form a biomimetic hybrid membrane in combination with photothermal therapy. Wang et al. [184] employed a hybrid membrane coated onto hollow polydopamine NPs in order to target melanoma homogeneously and bestow immunostimulatory qualities. The growth of melanoma could dramatically be inhibited, and the expression of IL-12P40 and IFN-γ cytokine was increased after tail vein administration. Such an excellent therapeutic effect was achieved through synergistic photothermal therapy and immunotherapy. Consistent with previous findings, it was found that NPs coated with cancer cell membranes have specific target homing effects in this study, thus providing new insights and solutions for tumor-specific-targeted therapy. Similarly, combined with PTT, according to the study by Gao et al. [128], it was reported to use an ultrasonic bath to coat the surface of gold nanoparticles (GNPs) with a bacterial biomimetic membrane derived from *E. coli* (ATCC 33694), and then the mixture was centrifuged to precipitate the successfully coated biomimetic GNPs. Bacterial OMVs are phagocytosed by macrophages and neutrophils due to the presence of PAMPs [23,122]. Bacterial OMV-modified nanocarriers are then transported to the tumor site by a form of hitchhiking on phagocytosis. Therefore, the uptake of GNPs in macrophages RAW-264.7 was enhanced through the coating of bacterial OMVs. The intracellular pro-inflammatory cytokine TNF-α, IL-1β, and M1 polarizations were upregulated in RAW-264.7 cells. Subsequently, bacterial membrane biomimetic GNPs recruited immune cells in the tumor sites combined with PTT treatment in vivo, leading to the upregulation of inflammatory cytokines, the polarization of M1 macrophages, and the recruitment of DC cells. Similarly, the presentation of T-cell antigens in turn brought fantastic melanoma (B16) tumor regression in vivo. 

Unlike the use of macrophages as a medium for drug hitchhiking [128], a recent study [127] developed *E. coli* (ATCC 25922) bacterial OMV-modified multifunctional Fe_3_O_4_-MnO_2_ nanocarriers (FMO) through ultrasonic fusion to enhance tumor immunotherapy under NIR laser irradiation. FMO NPs were phagocytosed by neutrophils and targeted to solid tumor sites in B16F10 tumor-bearing mice and then induced B-cell and T-cell special antitumor immune responses. The synergistic nanoplatforms recruited more DC cells to the cancer location and upregulated antitumor pro-inflammatory cytokines, including TNF-α, IL-4, and IL-6, under NIR exposure, resulting in systematic antitumor effects and promoted the significant ablation of primary and distant tumors. Chen and colleagues [195] discussed the use of bacterial membrane vesicles (BMVs) coated with mesoporous polydopamine (MPD) nanoparticles via ultrasonic fusion as a potential new therapy for cancer treatment. The MPD@DMV formulation was evaluated for its ability to promote immune responses in melanoma tumor-bearing mouse models. Results showed that this formulation effectively upregulated T-cell infiltration and antitumor cytokines, leading to tumor regression and extended survival time in mice. Moreover, intravenous injection of MPD@DMV was found to have better long-term immune effects than intratumoral injection. The authors propose that this innovative formulation could serve as a foundation for future studies exploring the potential of combining bacteria-derived products with environmentally friendly materials for cancer therapeutics.

In a word, it is possible to utilize bacterial OMVs coated on the surface of nanomaterials via relatively easier and faster ultrasonic fusion, which is suitable for large-scale production and could reduce the loss of OMVs and NPs. However, the application is limited due to the fact that ultrasound may denature the membrane proteins. Significantly, the uneven distribution and uniformity of bacterial OMV-coated NPs remains to be solved via ultrasonic fusion.

#### 5.1.3. Other Bind Forms

Although membrane extrusion and ultrasonic fusion are most commonly employed to cover the biomimetic membrane on the surface of nanomaterials; these two types of cell membrane fusion might affect membrane topology and destroy the biofunctions of biomimetic nanocarriers [196]. Therefore, microfluidic electroporation, which is similar to ultrasonic incubation and has high reproducibility, could be used to wrap cell membranes onto nanoparticles [178]. Unfortunately, due to the expensive cost and the difficulties of large-scale preparation, few studies have been conducted to coat bacterial OMVs onto NPs with microfluidic electroporation. But microfluidic electroporation may be employed to fuse OMVs onto NPs in the near future [177]. Unlike the above studies, Qin et al. [187] constructed CuS-OMV binding sites by the biomineralization of CuS nanocrystals through the precipitation reaction of Cu^2+^ and S^2−^, binding to the proteins of OMVs as a nanoreactor template via the one-pot method. CuS-OMVs could efficiently target the solid tumor sites of 4T1 tumor-bearing mice upon NIR-II light irradiation. OMVs acted as immune adjuvants to induce DC maturation, enhance tumor infiltration, and activate CD8^+^ T cells. Meanwhile, CuS-OMVs convert the M2-like tumor-associated macrophages to the M1-like phenotype, significantly inducing immunogenic cell death. This brought up the remarkable suppression of primary and distant tumor growth.

### 5.2. Bacterial Ghost (BG) Nanoparticle-Based Nanoplatforms

BGs are intact bacterial membranes that have similar structural functions to bacterial OMVs. BGs are also capable of activating the internal and adaptive immune response. Therefore, BGs are often used as vaccine carriers. In one study, bacterial ghost-based nanoplatforms were developed [84], in which the chemotherapeutic drug 5-fluorouracil (FU) and the macrophage phenotype modulator zoledronic acid (ZOL) were loaded into the facultative anaerobic probiotic *E. coli* Nissle 1917 via electroporation. Au nanorods were loaded on the surface of EcN. The active EcN moved into the hypoxic TME by means of its inherent tropism; EcN Z/F@Au transformed into BGs under near-infrared (NIR) conditions and then gradually turned on drug release under the spatiotemporal level. ZOL transformed macrophages M2 into M1 macrophages when EcN Z/F@Au was directed into the tumor tissue. The BG-based nanocarriers effectively stimulated the immune responses and increased the expression of pro-inflammatory cytokines to synergistically inhibit the proliferation and growth of the 4T1 tumor. This study combined chemotherapy, immunotherapy, and photothermal therapy and maximized the antitumor effect with few serious side effects. Thus, BG-based nanocarriers could provide a distinctive and personalized tumor-targeting therapy idea for the current dilemma in the antitumor research field.

### 5.3. Bacterial Spore–Nanoparticle-Based Nanoplatforms

Probiotic spores have been employed to treat clinic tumor patients, and have received prospective therapeutic results [41]. The strategy of combining NPs and spores may produce more valid therapeutic efficiency for cancer biotherapy. For instance, *Clostridium novyi-NT* spores were encapsulated around branch gold NPs through simple electrostatic deposition [91], which possessed the capability of reducing systemic toxicity and enhancing the anticancer efficacy. Meanwhile, the combination of spores and gold NPs produced a modest influence on ability of living spores to proliferate into the tumor microenvironment. The process of delivering the living spore-based nanoplatforms to the tumor site could be visualized by computed tomography. A similar study about anaerobic *Clostridium novy-NT* spores for the treatment of glioblastoma with positively charged metformin-loaded peptide hydrogel (MRM) bound to the surface of spores negatively charged via electrostatic interactions was constructed [197]. The most effective tumor suppression effects on GL261-bearing C57BL/6 mice and the survival time were greatly prolonged by the administration of MRM-coated spores. Meanwhile, it can reshape the immune microenvironment of glioma, promote the maturation of DC cells, significantly upregulate the proportion of NK cells, and activate the adaptive immune response caused by T lymphocytes. Macrophages have a decisive role in regulating the TME [23]. The treatment group of MRM-coated spores could also effectively upregulate the proportion of M1 macrophages with antitumor effects and induce the polarization of M1 macrophages. Additionally, the researchers discovered that MRM-coated spores triggered immunological memory effects, which dramatically inhibited the growth of glioblastoma in GL261-bearing C57BL/6 mice inoculated again. Probiotic spores have been widely employed as oral agents to regulate immune levels and treat malignant tumors [89,153,198]. 

An oral drug delivery system consisting of *Clostridium butyricum* spores covalently linked with mesoporous silica nanoparticles loaded with gemcitabine (MGEM) was designed to actively target to the locations of PDAC tumor under oral administration in PDAC tumor-bearing mice [198]. Compared with injection of single MGEM NPs, spore-MGEM could enrich intratumoral drugs about 3-fold in pancreatic tumor sites. Moreover, the oral drug delivery systems exhibited significant tumor regression in Panc-02 and Panc-01 tumor-bearing mouse models without significantly adverse effects on biosafety. Song et al. [89] constructed another oral drug delivery nanoplatform consisting of *Bacillus cagulans* spores decorated with deoxycholic acid (DA) and doxorubicin/sorafenib (DOX/SOR) through electrostatic interactions for cancer therapy. Interestingly, when DOX/SOR/Spore-DA moved into intestinal microenvironment, the ligand and chemotherapeutic medicines could autonomously assemble nanoparticles in vivo and in vitro, and spores germinated to active probiotics to colonize in tumor sites. Therefore, modified spores could protect the medicines from rugged regions of stomach and be beneficial to transport drugs to intestinal environment under oral administration. In another study, a *Bacillus* spore-based nanocarrier linking covalently with curcumin and folate was developed [153]. Consisting with the research by Song’s group, spores delivered curcumin complexes to colon tumor sites via the function of colon targeting after oral administration. Moreover, curcumin complexes assembled as nanomicelles in intestinal mucosa regulated the Caspase-3 mediated signaling pathway and resulted in tumor cell apoptosis. In all, the advantages of probiotic spores, including safety, low cost, high drug loading, and tumor-targeting ability, promoted the combination with NPs for cancer therapy.

### 5.4. Other Bacterial Component-Based Nanoplatforms

Bacterial components combined with nanotechnology for antitumor therapy are not only limited to bacterial OMVs, BGs, and BSPs, but also include bacterial proteins. The component equipped with specific functions could modify synthetic NPs and also achieve better biological activity. Dong et al. [199] proposed the opca protein of *Neisseria* through prokaryotic expression technology. Opca protein was bound to the surface of MnO_2_ synthetic NPs loaded with chemotherapeutic drug methotrexate MTX via bioconjugate chemistry methods. The presence of opca protein may dramatically reduce the pathogenic undesirable impacts caused by bacteria while imitating *Neisseria meningitidis* to overcome the blood–brain barrier (BBB). The authors found that biomimetic nanosystems could simultaneously improve the effectiveness of chemotherapeutic agents to combat tumors via reducing the anaerobic environment of tumor tissue and decomposing H_2_O_2_ to release oxygen. The *E. coli* membrane protein and adhesion protein FimH were recently combined with Au nanorods (AuNRs) by ultrasonication [200]. It was possible to diminish the toxicity of AuNR brought on by CTAB by surface modification. Additionally, LPS was eliminated utilizing ClearColi lysate, eliminating the risk of sepsis. Subsequently, FimH caused the activation of DC cells, T cells, and NK cells. Under laser irradiation, the synthesized ECA caused tumor cells at the 4T1 and CT-26 tumor sites to undergo apoptosis and necrosis. In the meantime, ECA raised IL-6 and IL-12p40 expression levels. In short, the binding of immunogenic bacterial fractions with nanoparticles could produce strong therapeutic effects on primary tumors and prevented tumor recurrence and metastasis.

## 6. Future and Clinical Trials of Bacteria- and Bacterial Component-Based Nanoplatforms in Cancer Therapy

Bacteria and bacterial component-mediated cancer therapy have raised increasing attention and numerous clinical trials have been conducted. Currently, there are 465 clinical trials on *E. coli* (104), *Salmonella* (9), *Listeria* (21), *Lactobacillus* (75), *Bifidobacterium* (42), *Bacillus* (133), and *Clostridium* (81) alone, and 142 trials have entered phase Ⅲ/Ⅳ in the clinic (as of May 2023). Representative examples of bacteria- and bacterial component-mediated cancer therapy in clinical studies are summarized in Table 5 from https://clinicaltrials.gov/ (accessed on 15 May 2023). In fact, bacterial-mediated tumor therapy has been studied relatively early. In 1893, Dr. Coley treated several tumor patients employing heat-inactivated *Streptococcus pyogenes* and *Serratia marcescens* administered into the tumor sites, and eventually healed some of them [201]. However, the mechanism of tumor regression was not clearly investigated at that time. Later, phase I clinical trials suggested that bacterial toxins resulted in the upregulation of immune cytokine expression in patients such as TNF-α, IFN-γ, and IL-1β, which may bring about effective anticancer effects (NCT00623831) [33]. 

*Bacillus Calmette-Guerin* (BCG) has been shown to be successful in tumor therapy, and the FDA has approved BCG for the treatment of bladder cancer [20,34]. BCG is commonly utilized as bacterial tumor vaccine. Currently, 243 clinical trials involving the therapeutic use of BCG in oncology have been conducted, 67 of which have entered the clinic’s phase Ⅲ/Ⅳ. In another clinical trial, Axalimogene filolisbac (ADXS-HPV), a living attenuated *Listeria monocytogenes*-based vaccine, has been evaluated in phase Ⅱ clinical trials for cancer therapeutics (NCT02002182). The outcome demonstrated that ADXS-HPV was efficient and consistent with protocol requirements; however, further clinical trials are still required [202]. Although it has been proven that attenuated *Salmonella* could aggressively target tumor regions and result in tumor regression in mice, the administration of a dosage of the live attenuated VNP20009 produced some bacterial tumor colonization in only two individuals and no obvious tumor regression in a clinical trial phase I including 24 patients (NCT00004988). Fortunately, the injection of VNP20009 did not raise any biosafety issues. Moreover, VNP20009 increased the expression of IL-1β, TNF-α, IL-6 and IL-12, all of which are pro-inflammatory cytokines [35]. Such regrettable outcomes could be the result of genetic engineering interventions that remove bacterial virulence genes, inducing an inadequate immune response to fight the tumor [20].

Attenuated Salmonella expressing the *E. coli* cytosine deaminase gene could convert 5-fluorocytosine (5-FC) to 5-fluorouracil (5-FU). The findings of the trial showed that the gene-delivery-engineered attenuated bacteria were able to colonize some of the patients’ solid tumor sites, considerably enhancing the expression levels of 5-FU/plasma. Three administration cycles considerably increased the survival time of patients [203]. In another clinical phase I study, attenuated *Salmonella typhimurium* that contained the human gene for IL-2 could effectively activate NK cells and NK-T cell immune response in the bodies of 22 patients with metastatic gastrointestinal cancer via oral administration (NCT04589234). Although the trial indicated no appreciable adverse effects after oral administration, further research at various doses is still required [204]. 

In addition to living bacteria as drug delivery systems to treat solid tumors, bacterial components also showed the excellent ability of immune adjuvants and are highly effective anticancer treatments. For example, researchers [125] used EGFR-EDV-Dox as a drug delivery system to conduct a phase I clinical trial (NCT02766699) on pancreatic ductal adeno-carcinoma and Glioblastoma Multiforme (GBM). The antitumor immune response promoted tumor regression, resulting in advances in bacterial component-based clinical oncology. Similarly, bacterial outer membrane protein A derived from *Klebsiella pneumoniae* acted as Toll-like receptor 2 ligand P40. In a phase I clinical trial, Lienard et al. [94] utilized bacterial outer membrane protein as a vaccine adjuvant, along with tumor antigen, to inoculate 14 melanoma patients. Lastly, noteworthy observations were made, with certain patients demonstrating distinct T-cell responses following vaccination. Moreover, a subset of these patients undergoing treatment exhibited augmented expression of IFN-γ, signifying its production by T cells, and 3/14 patients achieved full tumor remission. In another phase III clinical trial [95], a cancer vaccine containing the *Neisseria meningitidis* outer membrane protein complex was able to elicit an inflammatory response in the participants and facilitate the induction of tumor remission in synergy with chemotherapeutic agents, but the precise mechanism of the T-cell response remains to be determined in additional clinical trials. Bacterial spores, as common bacterial components with tumor-targeting ability, are usually utilized to treat solid tumors, and achieved the desired efficacy in clinical trials. The first human injection of *Clostridium novyi-NT* spores in solid tumors was conducted in a phase I clinical trial by Janku et al. (NCT01924689) [41]. Following injection of a specific dose of bacterial spores, tumor cell lysis was observed in 10 of the 24 patients (42%) with solid tumors. It was determined that *C. novyi-NT* injection would trigger a systemic immune response and boost the tumor cell-specific T-cell activation. 

Bacteria and bacterial components have the potential to be used as therapeutic delivery vehicles due to their unique characteristics [10,20,42,122,205]. A multitude of clinical trials have indicated that genetically/bioengineered bacteria and bacterial derivatives exhibit minimal adverse effects [35,202,203]. Numerous preclinical studies focusing on bacterial and bacteria-derived drug delivery systems have presented promising results, showcasing their potential in inducing tumor regression [39,41,97,114,118]. Furthermore, bacteria-mediated cancer therapy has been combined with chemotherapy, photodynamic therapy, photothermal therapy, radiotherapy, and nanotechnology to obtain greater synergistic antitumor effects in mice in multiple preclinical investigations. This strategy could be applied to speed up the clinical translation of bacterial and bacterial component-based drug delivery systems in prospective clinical trials.

## 7. Safety Issues of Bacteria and Bacterial Components

Despite the advantages of bacteria and bacterial components for cancer therapy, safety is still a major concern. Even the FDA-approved *Bacillus Calmette-Guérin* (BCG) vaccine for treating bladder cancer presents some side effects, including inflammation, bacterial infections, and sepsis [25,108]. Firstly, numerous bacteria have pathogenicity with strong colonization capabilities [206]. The pathogenicity of bacteria introduces risks to biological organisms. Secondly, most bacteria possess strong immune activation capacity, which may result in adverse effects, such as hemolysis and thrombosis [20]. Thirdly, the excessive inflammation associated with PAMPs involved in the recognition of bacteria or bacterial components may trigger a cytokine syndrome when entering the body’s circulation [207]. This phenomenon represents an overactivation of the immune system, which could result in uncontrolled inflammatory responses, causing damage to normal tissues and organ functions. Lastly, many bacteria are carcinogenic, which could release undesired substances and accelerate cancer progression [20].

Currently, there are two main approaches to enhancing bacterial safety through engineering methods, involving the reduction of virulence and the augmentation of tumor targeting [109,110,208]. Genetic engineering is employed to eliminate or weaken virulence genes, which is primarily achieved through the modification of LPS, significantly enhancing the safety of bacterial therapy [108]. However, some virulence genes may contribute to antitumor immune activity, necessitating the maintenance of their antitumor immune activity while attenuating their toxicity [139,209]. Nevertheless, precise strategies for manipulating gene expression within bacteria are currently lacking [156]. Moreover, strategies for enhancing the tumor specificity of bacterial therapy include constructing nutrient-deficient bacteria or attaching adhesive peptides [109,110,208], tumor-related antigens, and tumor-targeting ligands to engineered bacteria and bacterial components. Furthermore, intelligent nanomaterials and advanced nanotechnologies are utilized to facilitate bacterial aggregation at tumor sites, achieving the precision of drug release [146,156]. This has promoted the development of a bacteria/bacterial component–nanoparticle hybrid platform that is much safer and more efficient for cancer therapy.

## 8. Conclusions

Bacteria and bacterial components offer promising prospects as natural bio-nanocarriers due to their inherent tumor-targeting, easy design and modification, intrinsic immunostimulatory characteristics, and high drug delivery efficiency. Bacteria are more prone to accumulate and proliferate in anaerobic tumor areas, and most bacteria and bacterial components can be cleared by the immune system within normal healthy tissues. Moreover, bacteria and their components inherently possess the capacity to interact with immune cells and stimulate the immune system to kill tumor cells, and have the potential to be effective against MDR tumor cells. Furthermore, clinical trials have shown that bacteria and bacterial components could activate the innate immune system, and engineered bacteria and derivatives have promoted tumor regression. However, the potential safety issues, low therapeutic efficacy, uncontrollable drug-loaded concentration, and ambiguous mechanism significantly limit the clinical transformation. 

In all, chemical and biological engineering strategies could be employed, along with nanotechnology, to improve biosafety and achieve synergistic therapy. Many studies have utilized chemical binding, genetic engineering, biomimetic surface coating, and surface modification to enhance the therapeutic effects and improve the biosafety of bacteria-based delivery systems. Moreover, nanoparticles could bind specifically to engineered bacteria through chemical bonds, electrostatic interactions, biotin–streptavidin, and electroporation, and they also could fuse with bacterial components via membrane extrusion, ultrasonic fusion, and microfluidic electroporation. Such biomimetic nanocomplexes could significantly improve the multifunctional effectiveness of cancer therapy and become a potential strategy to eradicate tumors. Bacteria and bacterial components, as naturally biological nanocarriers combined with engineered strategies and nanotechnology (Figure 9), are expected to become a more formidable and innovative frontier weapon against tumor progression in near future.

## Figures and Tables

**Figure 1 pharmaceutics-15-02490-f001:**
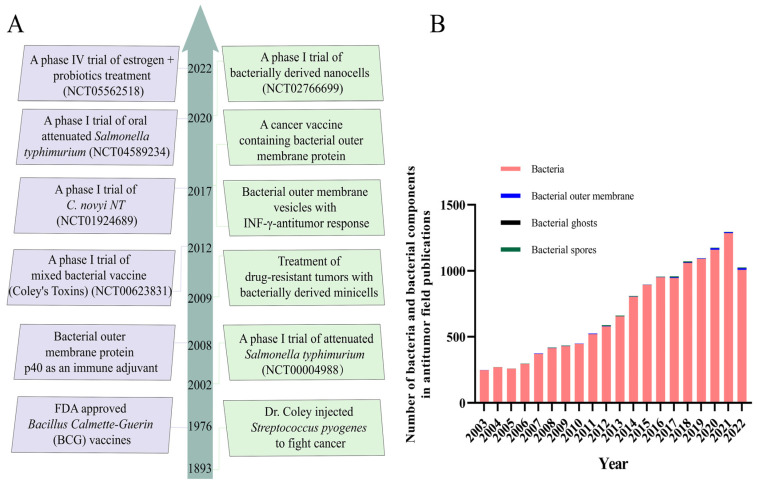
(**A**) Timeline of representative cases of bacteria-/bacterial component-mediated cancer therapy; (**B**) number of bacteria and bacterial components in antitumor field publications from PubMed in recent 20 years (2003–2022).

**Figure 2 pharmaceutics-15-02490-f002:**
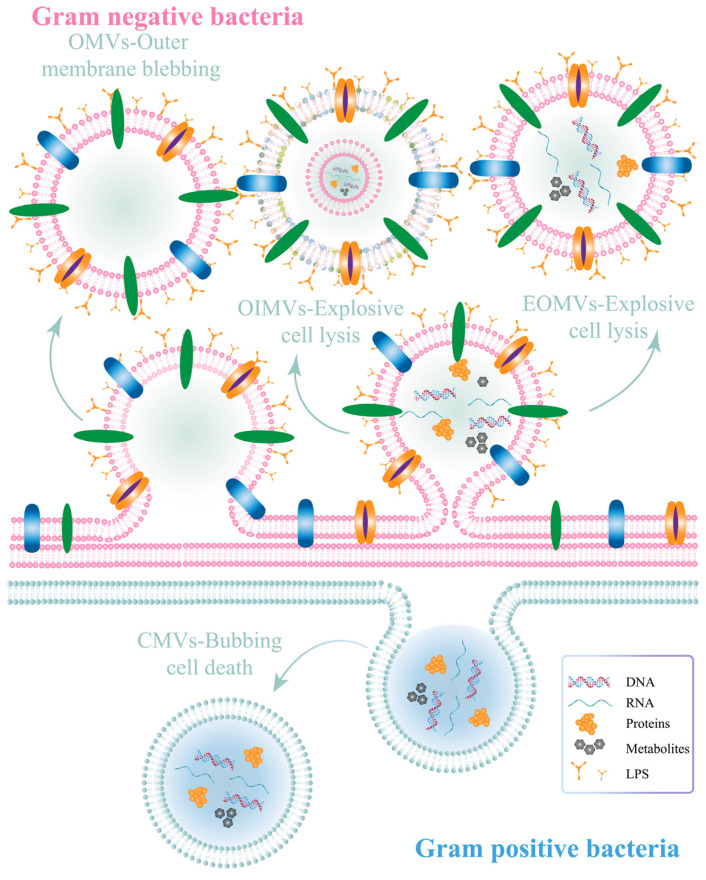
Biogenesis and composition of bacterial OMVs. EOMVs—explosive outer membrane vesicles; OIMVs—outer inner membrane vesicles; CMVs—cytoplasmic membrane vesicles.

**Figure 3 pharmaceutics-15-02490-f003:**
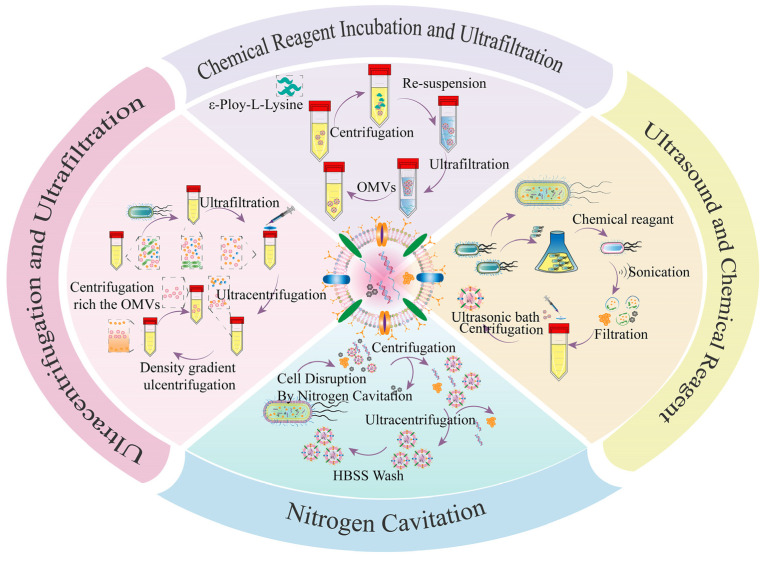
Isolation and purification of bacterial OMVs by ultracentrifugation, ultrafiltration, sonication, and nitrogen cavitation.

**Figure 4 pharmaceutics-15-02490-f004:**
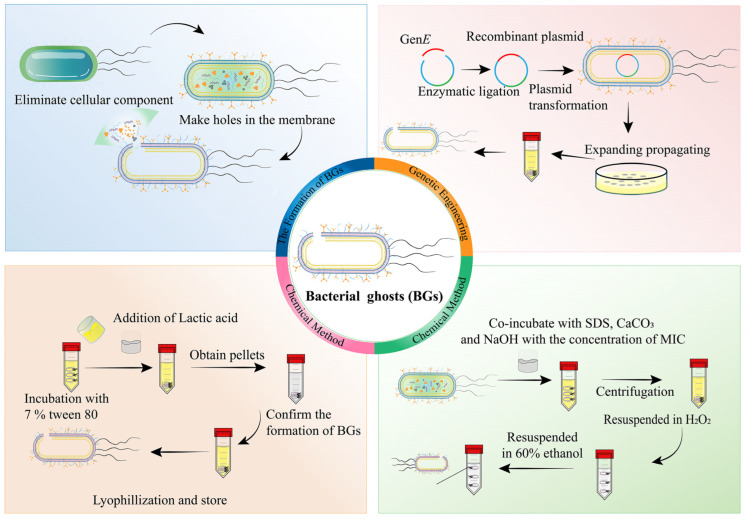
Structure, composition, isolation, and purification of bacterial ghosts.

**Figure 5 pharmaceutics-15-02490-f005:**
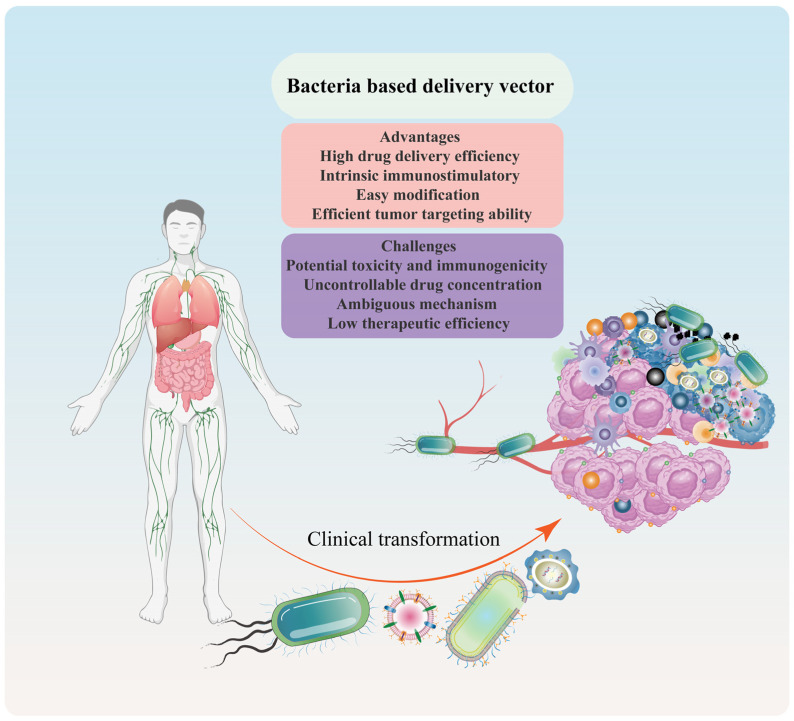
Advantages and challenges of bacteria-based delivery vector in clinical transformation.

**Figure 6 pharmaceutics-15-02490-f006:**
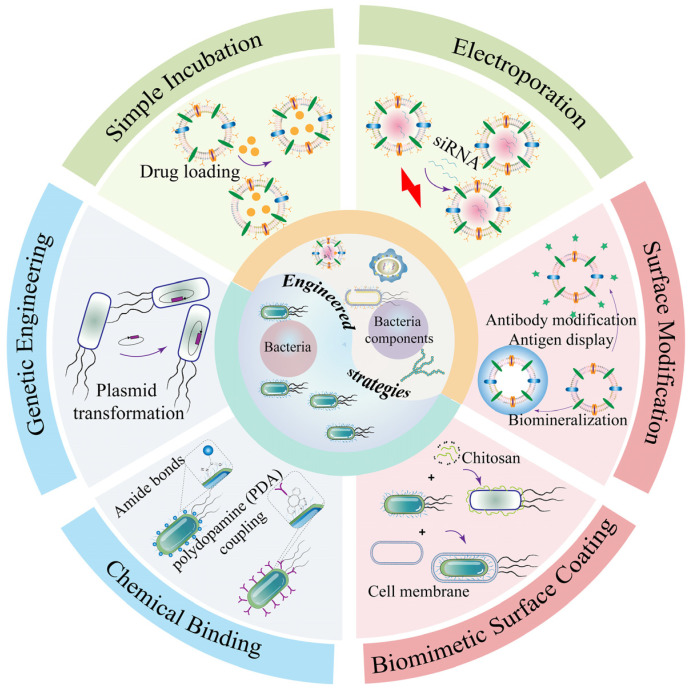
Schematic illustration of engineered strategies of bacteria/bacterial components for drug and gene delivery in cancer therapy.

**Figure 7 pharmaceutics-15-02490-f007:**
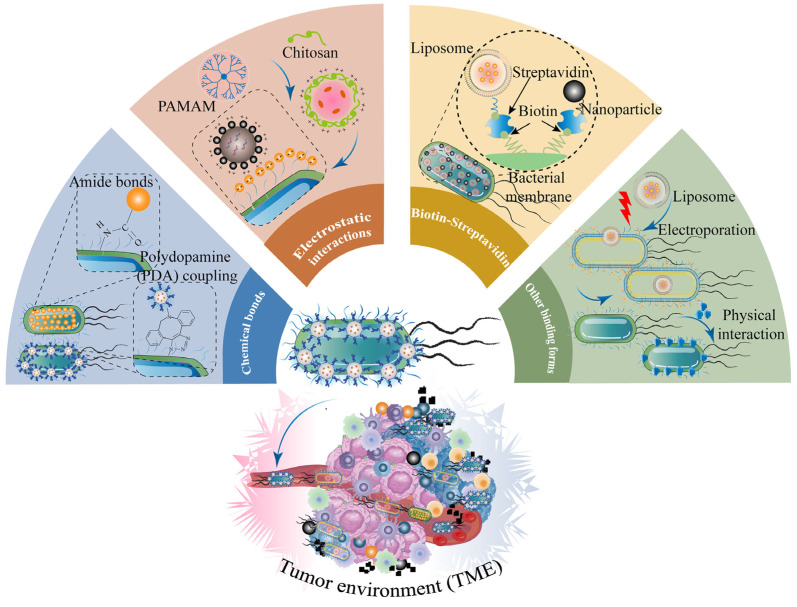
Schematic illustration of the fabrication of bacteria nanoparticles via chemical bonds, electrostatic interactions, biotin–streptavidin, and other binding forms.

**Figure 8 pharmaceutics-15-02490-f008:**
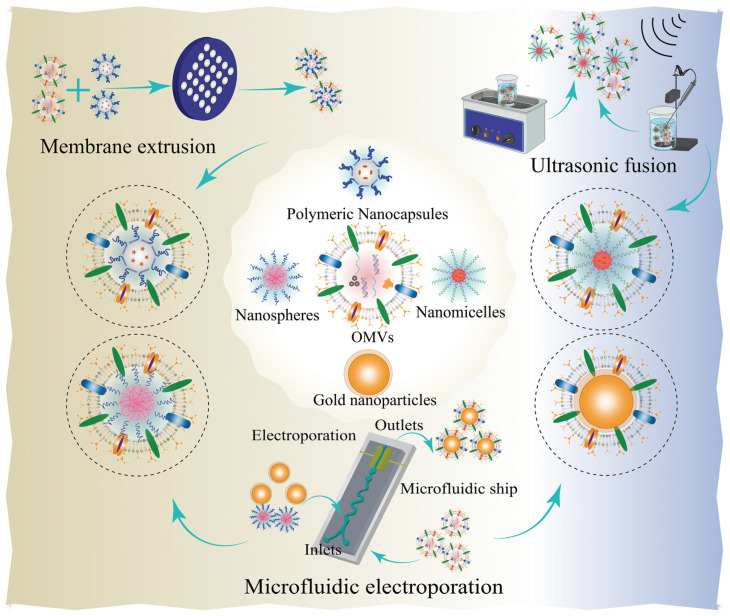
Schematic illustration of the fabrication of bacterial OMV nanoparticles via membrane extrusion, ultrasonic fusion, and microfluidic electroporation.

**Figure 9 pharmaceutics-15-02490-f009:**
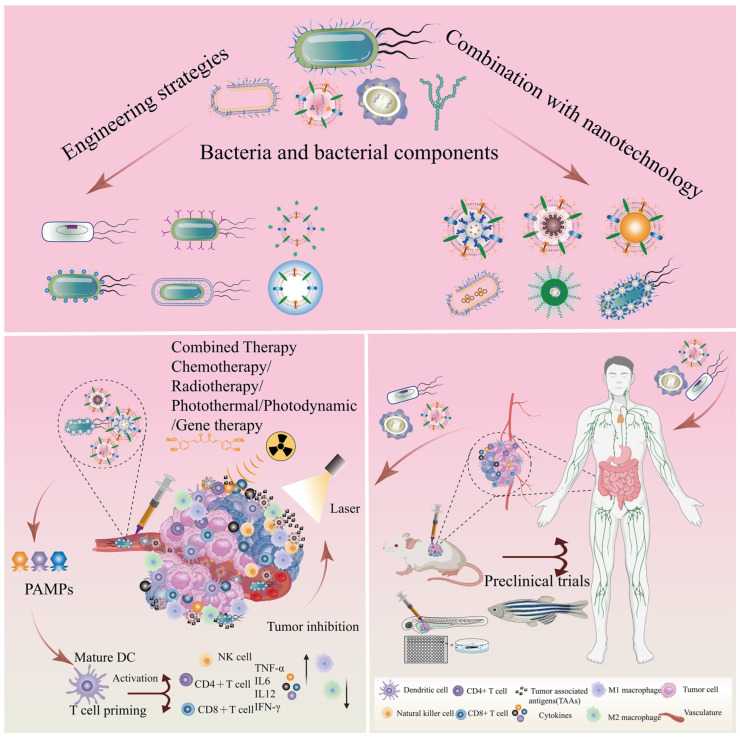
Bacteria- and bacterial component-based nanocarriers combined with engineered strategies and nanotechnology and the mechanism of cancer therapy (↑: up-regulate, ↓: down-regulate).

**Table 1 pharmaceutics-15-02490-t001:** Summary of recent research on bacteria and bacterial component-based drug delivery vectors for tumor therapy.

Bacteria/ Bacterial Components	Microorganism	Method	Results	Ref.
Bacteria	*E. coli* Nissle 1917 *Salmonella typhimurium* VNP20009	Chemical binding	Activated immune responses Targeted intratumoral localication	[38]
Bacteria	*E. coli* BL21(DE3)	Chemical binding	Dual ability of tumor immune activation	[97]
Bacteria	*E. coli* MG1655	Genetic engineering	Actively targeted to solid tumor regions Induced tumor cell autophagy	[98]
Bacteria	*E. coli* Nissle 1917	Genetic engineering Biomimetic surface coating	Improved antitumor efficacy in vivo Increased microbial translocation in distal tumors	[99]
Bacteria	*Attenuated Salmonella typhimurium* VNP20009	Biomimetic surface coating	Synergistic and systematic antitumor immune responses Inhibited tumor progression and metastasis	[100]
OMVs	*E. coli* BL21 (DE3)	Genetic engineering Surface modification	Remodeled TME Long-time adaptive immune response	[23]
OMVs	*E. coli* DH5α	Genetic engineering	Inhibited tumor angiogenesis Promoted tumor cell apoptosis	[68]
OMVs	*Salmonella typhimurium* ATCC 14028	Simple incubation	Enhanced autophagy and apoptosis of tumor cells	[101]
OMVs	Attenuated *K. pneumonia* ACCC 60095	Simple incubation	Recruited macrophages in TME Promoted tumor cell apoptosis	[102]
BGs	*E. coli* Nissle 1917	Simple incubation	Promoted DC maturation Increased CD4^+^ and CD8^+^ T-cell proliferation	[103]
BGs	*E. coli* Nissle 1917	Simple incubation	Exhibited synergistic antitumor activity Induced immunogenic cell death	[104]
Spores	*Clostridium novyi-NT*	Simple incubation	Induced a systemic immune cytokine response Enhanced the tumor cell-specific T-cell activation	[41]
Spores	*C. butyricum* ATCC 19398	Simple incubation	Targeted and enriched in tumor sites	[93]

**Table 2 pharmaceutics-15-02490-t002:** Summary of recent research on bacteria-/bacterial component-based nanocarriers combined with nanoparticles for gene therapy.

Bacteria/ Bacterial Component	Microorganism	Gene	Type of Ligand	Method	Ref.
Bacteria	*Streptococcus mutans* (*S. m*)	ssDNAs	Nucleic aptamer AS1411	Surface modification	[143]
OMVs	*E. coli* BL21 (Δ*msbB*)	*Redd1* siRNA	DSPE -PEG	Electroporation Surface modification	[106]
OMVs	*E. coli* BL21 (DE3)	Box C/D mRNA	--	Genetic engineering Surface modification	[59]
OMVs	Engineered K-12 W3110 *E. coli*	*KSP* siRNA	HER2	Genetic engineering Biotin–streptavidin	[107]
OMVs	Engineered K-12 W3110 *E. coli*	PD-1 pDNA	LyP1	Genetic engineering Plasmid transfection	[144]
OMVs	Attenuated *Salmonella*	*CD38* siRNA	--	Ultrasonic fusion	[120]
Minicells	*Salmonella enterica serovar Typhimurium*	*PLK* siRNA *KSP* siRNA *MDR1* siRNA	EGFR	Simple incubation	[145]

**Table 3 pharmaceutics-15-02490-t003:** Summary of recent research on bacteria-/bacterial component-based nanocarriers combined with nanoparticles for tumor therapy.

Bacteria/ Bacterial Components	Microorganism	Nanoparticles	Method	Results	Ref.
Bacteria	*E. coli* Nissle 1917	Magnetic nanoparticles	Chemical bonds Genetic engineering Surface modification	Triggered with magnetothermal ablation NDH-2-induced ROS damage	[147]
Bacteria	*E. coli* MG1655	Magnetic nanoparticles Nanoliposomes	Streptavidin–biotin	Moved through the tumor spheroids autonomously under magnetic field	[148]
Bacteria	*E. coli* MG 1655	Nanocapsules	Chemical bonds	Colonized to the tumor sites Enhanced T-cell infiltration	[149]
Bacteria	*Salmonella typhimurium* VNP20009	Polyamidoamine dendrimer	Electrostatic interactions Biospecific binding	Enhanced antitumor immune responses	[150]
Bacteria	Attenuated *Salmonella* *typhimurium* (YS1646)	Liposomes	Streptavidin–biotin	Enhanced immune cell infiltration Excellent tumor-suppressive effects	[151]
OMVs	Attenuated *Salmonella typhimurium*	Nanomicelles	Membrane extrusion	Activated macrophages for the stimulation of T cells Prevented tumor metastasis	[152]
OMVs	*E. coli* K1	PLGA nanoparticles	Membrane extrusion	Prolonged the elimination of drugs Superior brain-targeting ability	[123]
OMVs	*E. coli* ATCC 25922	Fe_3_O_4_-MnO_2_ nanoparticles	Ultrasonic fusion	Targeted to solid tumor sites Induced antitumor immune responses	[127]
BGs	*E. coli* Nissle 1917	Au nanorods	Electroporation Physical adsorption	Stimulated the immune response Synergistic tumor inhibition efficacy	[84]
BSPs	*Bacillus coagulans*	Nanomicelles	Chemical bonds	Targeted to tumor sites Resulted in tumor cell apoptosis	[153]

**Table 4 pharmaceutics-15-02490-t004:** Advantages and challenges of bacteria and bacterial component-based nanocarriers combined with nanoparticles for tumor treatment.

	Method	Advantages	Challenges	Refs.
Bacteria	Chemical bonds	Strong bond association High spatiotemporal control	Modification of limited ligands Unavoidable bacteria damage	[29,147,155,158]
	Electrostatic interactions	Easy formation Multifunctional therapeutics	Poor stability of assemble conjugations	[29,166,167]
	Biotin–streptavidin	High binding affinity Better therapeutic effect	--	[29,151,172]
	Electroporation	Highly efficient anticancer effects High accumulative distribution	Side effect of viability of bacteria	[175]
Bacterial components	Membrane extrusion	Uniform size Better preservation of biomolecules	Time-consuming Difficult for large-scale production	[176,177]
	Ultrasonic fusion	Safe and non-toxic Faster and easier to perform Reduced loss of material	Denaturation of membrane proteins Drug leakage Lack of uniformity	[176,177,178]
	Microfluidic electroporation	Accurated control of size High reproducibility	Not commercially available Need to explore the scalability	[176,178]

**Table 5 pharmaceutics-15-02490-t005:** Examples of currently evaluated bacteria and bacterial components for antitumor therapy ^a^.

Bacteria /Bacterial Component	Clinical Trial Identifier	Cancer Types	Interventions	Status	Route
Bacteria	NCT05562518	Breast Cancer	Probiotics	Phase IV	Local administration
Bacteria	NCT04874883	Colorectal Cancer	Simbyotic	Phase IV	Oral administration
Bacteria	NCT03742596	Colorectal Cancer	Probiotic formula capsule	Phase II/Phase III	Oral administration
Bacteria	NCT01579591	Rectal Cancer	Probiotics	Phase III	Oral administration
Bacteria	NCT02002182	Squamous Cell Carcinoma	Modified *Listeria monocytogenes*	Phase II	Intravenous administrations
Bacteria	NCT03847519	Lung Cancer	Attenuated *Listeria monocytogenes*	Phase I/Phase II	Intravenous administrations
Bacteria	NCT01266460	Carcinoma	Attenuated live *Listeria* Encoding HPV 16 E7	Phase II	Intravenous administrations
Bacteria	NCT01099631	Liver Cancer Biliary Cancer	Biological: *Salmonella typhimurium*	Phase I	Oral administration
Bacteria	NCT00004988	Advanced or Metastatic Cancer	*Salmonella typhimurium* VNP20009	Phase II	Intravenous administrations
Bacteria	NCT04589234	Pancreatic Cancer	Salmonela-IL2	Phase II	Oral administration
Bacteria	NCT00623831	Malignancies	Mixed bacteria vaccine	Phase I	Subcutaneous administration
Bacterial Components	NCT02766699	Glioblastoma	Bacterially derived nonviable nanocells	Phase I	Intravenous administrations
Bacterial Components	NCT01924689	Solid Tumor Malignancies	*Clostridium novyi-NT* spores	Phase I	Intratumoral injection
Bacterial Components	NCT01118819	Solid Tumor Malignancies	*Clostridium novyi-NT* spores	Phase I	Intratumoral injection
Bacterial Components	NCT00358397	Tumors	*Clostridium novyi-NT* spores	Phase I	Intravenous administrations

^a^ Based on online information at the clinical trial website (https://clinicaltrials.gov/ (accessed on 15 May 2023)).

## Data Availability

Not applicable.

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
