# Peer review of "Bacteria and Bacterial Components as Natural Bio-Nanocarriers for Drug and Gene Delivery Systems in Cancer Therapy"

_pharmaceutics, 2023, doi:10.3390/pharmaceutics15102490_

Round 1
Reviewer 1 Report
Congratulations to the authors on a great review of high quality and rigor. The manuscript can be accepted after some only minor issues to be addressed, as summarized below:
* The authors mentioned risk of pathogenicity related to bacteria or bacteria components but did not further elaborate it more. The other related risk in my opinion may be over-inflammation associated with recognition of the bacteria or bacteria components, triggering cytokine release syndrome. The authors may consider having a paragraph (maybe towards the end of the manuscript) to discuss these aspects and what would be good engineering considerations to avoid such risks.
* It would be helpful to include the clinical trial numbers in Fig. 1A.
* Grammar error in line 108.
* Fig. 3 is somewhat cumbersome. I think it may not be necessary to include the exact experimental conditions (such as temperature etc.) in this figure, so it can be more clearly presented. Otherwise consider reworking it to make the major steps clear.
* Possible error with “mediated macrophages” on line 1063-1064.
Reviewer 2 Report
Dear authors,
I have several considerations about your review:
1. I kindkly ask you whether you use some AI software to condense references and write some parts of text? If so, please indicate it in your reply.
2. Figure 1A should be reconstructed because the size of letter is too small and it is not possible to read it. (May be in vertical?)
3. Correct in line 108, the word bacterial
4. It would be great if you introduce a list of acronyms at the end of manuscript: OMV, BG, BSP, TME, EPR and so on.
5. When you write Fig.3 and fig. 4 in lines 206 and 267, it should type Figure 3 and Figure 4 (full letters)
6. Figure 4 contains so small letter size. Please, do it bigger.
7. Line 285, reference 93 should be located at the end of the sentence, just after the word "membrane".
8. Figure 5 should appear just it is mentioned in the text, and not before.
9. In line 458, please change this part : "A recent work [114]...]" by "Li and co-workers...
10. The subheading 3.2 Bacterial components as nanocarriers for anti-cancer agents should be fully re-written. It is repetitive with previous information. From line 480 to line 523: What kind of anti-cancer agents are evolved??? There is not sense at all. Same for lines 543-545: "In another study, bacterial OMVs extracted from Salmonella effectively.....". What is the relation with anti cancer drugs???
11. From lines 568 to 598, I do not see the scientific sense.
12. Please, revise the further text to avoid unnecessary repetitions of qualitative text.
It is fine.
Reviewer 3 Report
In this review paper, the authors provide a comprehensive analysis of how bacteria and bacterial components can be used as anticancer therapeutic strategies. Starting from the limitations of anti-cancer strategies using nanocarriers, the authors summarize and explain the anti-cancer strategies using bacteria and bacterial components. In addition, from an "engineering" perspective, the authors reviewed bacteria-based modified anticancer therapies and combination treatment strategies and introduced ongoing/completed clinical trials. Overall, this reviewer agrees that it is a well-written and comprehensive review of this field. Although some of the sections are a bit long, the overall content covered is of interest to the reader. The following changes would make it easier for readers to understand.
1. The text in the figures is too small and difficult to read. In particular, Figures 1, 3, 4, and 7 have a lot of text in the figure, but it is too small and needs to be enlarged. In addition, it would be more helpful for readers if the full names of the abbreviations used in the figures themselves (Figure 2: OMV, OIMV, EOMV, etc.) were included in the figure legend so that the figures could stand alone.
2. It is recommended to add a section to explain the safety issues (e.g., bacterial infection or septic shock, cytokine storm, etc.) related to anticancer drugs using bacteria and bacterial components.
3. Sections 4.1-4.4 are long, but it is a little difficult to understand because there are no figures to help us understand. We recommend adding one figure to cover sections 4.1-4.4.
Round 2
Reviewer 2 Report
Dear authors,
Thank you so much for take in considerations my suggestions. I think you have improved enough the manuscript.
Congratulations!